# The phytochrome interacting proteins ERF55 and ERF58 repress light-induced seed germination in *Arabidopsis thaliana*

Zenglin Li[1], David J. Sheerin [1], Edda von Roepenack-Lahaye[2], Mark Stahl[2] & Andreas Hiltbrunner [1,3✉]

Seed germination is a critical step in the life cycle of plants controlled by the phytohormones abscisic acid (ABA) and gibberellin (GA), and by phytochromes, an important class of photoreceptors in plants. Here we show that light-dependent germination is enhanced in mutants deficient in the AP2/ERF transcription factors ERF55 and ERF58. Light-activated phytochromes repress *ERF55/ERF58* expression and directly bind ERF55/ERF58 to displace them from the promoter of *PIF1* and *SOM*, genes encoding transcriptional regulators that prevent the completion of germination. The same mechanism controls the expression of genes that encode ABA or GA metabolic enzymes to decrease levels of ABA and possibly increase levels of GA. Interestingly, *ERF55* and *ERF58* are themselves under transcriptional control of ABA and GA, suggesting that they are part of a self-reinforcing signalling loop which controls the completion of germination. Overall, we identified a role of ERF55/ERF58 in phytochrome-mediated regulation of germination completion.

[1] Institute of Biology II, Faculty of Biology, University of Freiburg, Freiburg, Germany. [2] Centre for Plant Molecular Biology, University of Tübingen, Tübingen, Germany. [3] Signalling Research Centres BIOSS and CIBSS, University of Freiburg, Freiburg, Germany. ✉email: andreas.hiltbrunner@biologie.uni-freiburg.de

Light is one of the most important environmental cues for plants as it has a strong impact on growth and development. Plants have evolved different classes of photoreceptors to sense and respond to changes in the environment. Phytochromes are photoreceptors that perceive red (R) and far-red (FR) light[1]. In the model plant *Arabidopsis thaliana*, phytochromes are represented by five members, of which two, phyA and phyB, have dominant functions. PhyA and phyB control the timing of developmental transitions such as seed germination, de-etiolation, and the induction of flowering, and play a key role in responses to canopy shade[1].

Seed germination is a critical step in the life cycle of plants. Seeds are relatively well protected against adverse environmental conditions, whereas seedlings emerging from germinated seeds are much more vulnerable[2]. Proper timing of the completion of germination is therefore an important trait in crops and crucial for plant survival in natural environments. The phytohormones abscisic acid (ABA) and gibberellin (GA) control seed germination in opposite directions, with ABA inhibiting and GA promoting the completion of germination[3]. ABA accumulates in seeds during seed maturation and is rapidly depleted during imbibition. ABA INSENSITIVE 5 (ABI5) and DELLAs, including GA INSENSITIVE (GAI), REPRESSOR OF GA1 (RGA), and RGA-LIKE 2 (RGL2), are important repressors of the completion of seed germination. ABI5 is a transcription factor that promotes ABA responses, whereas DELLAs inhibit GA signalling[3–6]. High GA levels induce the degradation of DELLAs and thereby promote the completion of seed germination[5,6].

In Arabidopsis and many other species, light controls the completion of seed germination through phyA and phyB[1,7]. The inactive state of phytochromes, Pr, has an absorption peak at 660 nm (R light), whereas the absorption peak of the physiologically active Pfr state is at 730 nm (FR light). Phytochromes reversibly convert between Pr and Pfr by absorption of light, with the relative proportion in the active state (Pfr/Ptot) being determined by the ratio of R and FR light in the environment[1,8]. PhyB is active in light conditions with a high R:FR ratio, such as monochromatic R light or sunlight, and inactivated by monochromatic FR light or canopy shade, where the R:FR ratio is low. In contrast, and for reasons that are not yet well understood, phyA is primarily active in FR light and low-light conditions, where relative Pfr levels are low[9].

Previous studies showed that phyA and phyB mediate light-regulated seed germination partly by binding and thereby destabilising the basic helix-loop-helix transcription factor PHYTOCHROME INTERACTING FACTOR 1 (PIF1)[7,10]. PIF1 directly associates with the promoter of *GAI*, *RGA*, and *ABI5* to induce their expression[11,12]. PIF1 also directly activates the transcription of *SOMNUS* (*SOM*), which enhances the expression of genes encoding GA catabolic or ABA anabolic enzymes; in contrast, the expression of genes encoding GA anabolic or ABA catabolic enzymes is repressed, thereby inhibiting the completion of germination[13–15]. However, there is no evidence for a direct regulation of genes encoding ABA or GA metabolic enzymes by PIF1 or SOM[7,12,13]. In addition to PIF1 and SOM, several other proteins play an important role in phyA and phyB regulated seed germination[3].

The AP2/ERF proteins represent a large transcription factor family in plants, with almost 150 members in Arabidopsis[16]. This family includes members that orchestrate various biological functions, such as flower development and responses to hormones or abiotic stresses[16–18]. Extensive redundancy probably exists within the AP2/ERF transcription factor family, and the genes whose expression AP2/ERFs control are often unknown.

ERF58 (ETHYLENE RESPONSE FACTOR 58) belongs to group A6 of the AP2/ERF transcription factor family, which consists of ERF53 to ERF62[16]. *ERF58* transcript levels are upregulated by ABA and heat stress; in vitro assays suggest ERF58 binds the dehydration responsive element (DRE; A/GCCGAC) and the coupling element CE1 (CCACC) within responsive promoters[19,20]. However, growth and development of *erf58* mutant plants does not differ from the wild-type and the function of endogenous *ERF58* is still unknown[19,21]. Overexpression of ERF58 leads to pleiotropic effects, including hypersensitivity to ABA during germination, reduced salt tolerance, and callus formation[19,21].

Here we show that ERF58 and its homologue ERF55 act redundantly during the phytochrome-controlled regulation of seed germination. Light-activated phyA and phyB repress the expression of *ERF55* and *ERF58*, and interact with the DNA-binding domain of ERF55 and ERF58, preventing their association with the promoter of target genes. This results in altered expression of genes encoding regulators of seed germination and thereby promotes the completion of germination.

## Results

### ERF58 interacts with phyA and phyB.
Evolution has shaped diverse seed germination strategies, with light serving as a germination trigger in positively photoblastic species. Light in the red and far-red spectral range is perceived by phytochromes. To identify phytochrome-dependent seed germination signalling pathways, we performed a yeast two-hybrid (Y2H) screen for proteins interacting with light-activated full-length phyA[22]. We reported previously on SPA1, PCH1, COR27, and NOT9B as phyA-binding proteins identified in this screen[22–25]. ERF58, a member of group A6 of the AP2/ERF transcription factor family[16], was another protein identified in the screen for phyA-interacting proteins. In addition to phyA, phyB also interacts with ERF58 in Y2H assays, and we observed interaction of phyA and phyB also with most other AP2/ERFs of group A6 in yeast, including ERF55 (Fig. 1a, b; Supplementary Fig. 1a). As *ERF58* and *ERF55* are transcribed in mature seeds in contrast to other members of the A6 AP2/ERF subfamily[17,26], this report focuses on ERF55 and ERF58. When co-expressed in *Nicotiana benthamiana* leaves, HA-YFP-ERF58 and HA-YFP-ERF55 co-localised with phyA-NLS-CFP and phyB-CFP in photobodies[27], indicating they may interact with phyA and phyB *in planta* (Fig. 1c; Supplementary Fig. 1b).

Next, we performed co-immunoprecipitation (Co-IP) assays using a stable transgenic line expressing *pERF58:HA-YFP-ERF58*. Seedlings were grown for three days in constant darkness and irradiated with different light conditions as indicated in Fig. 1d. Endogenous phyA and phyB co-immunoprecipitated with HA-YFP-ERF58 when activated by light (Fig. 1d), confirming that ERF58 is in complex with phyA and phyB *in planta*. We were not able to obtain stable transgenic lines expressing tagged ERF55 and therefore experiments depending on such lines are currently not possible. Collectively, our results show that ERF58 interacts with phyA and phyB in plants, and we have evidence that also other group A6 AP2/ERFs, including ERF55, bind phyA and phyB.

### ERF55 and ERF58 repress the completion of seed germination.
To characterise the physiological function of ERF55 and ERF58, and compare it to the function of other group A6 AP2/ERFs, germination phenotypes of two independent *ERF58* T-DNA insertion mutant alleles, *erf58-1* and *erf58-2* (Supplementary Fig. 2a, c), and one mutant allele for *ERF55*, *erf55-1* (Supplementary Fig. 2b, d), *ERF56*, *ERF57*, *ERF59*, and *ERF60* were examined. In addition, we examined germination also in ERF58ox lines, expressing HA-YFP-ERF58 under the strong, constitutive CaMV *35S* promoter (Supplementary Fig. 2e).

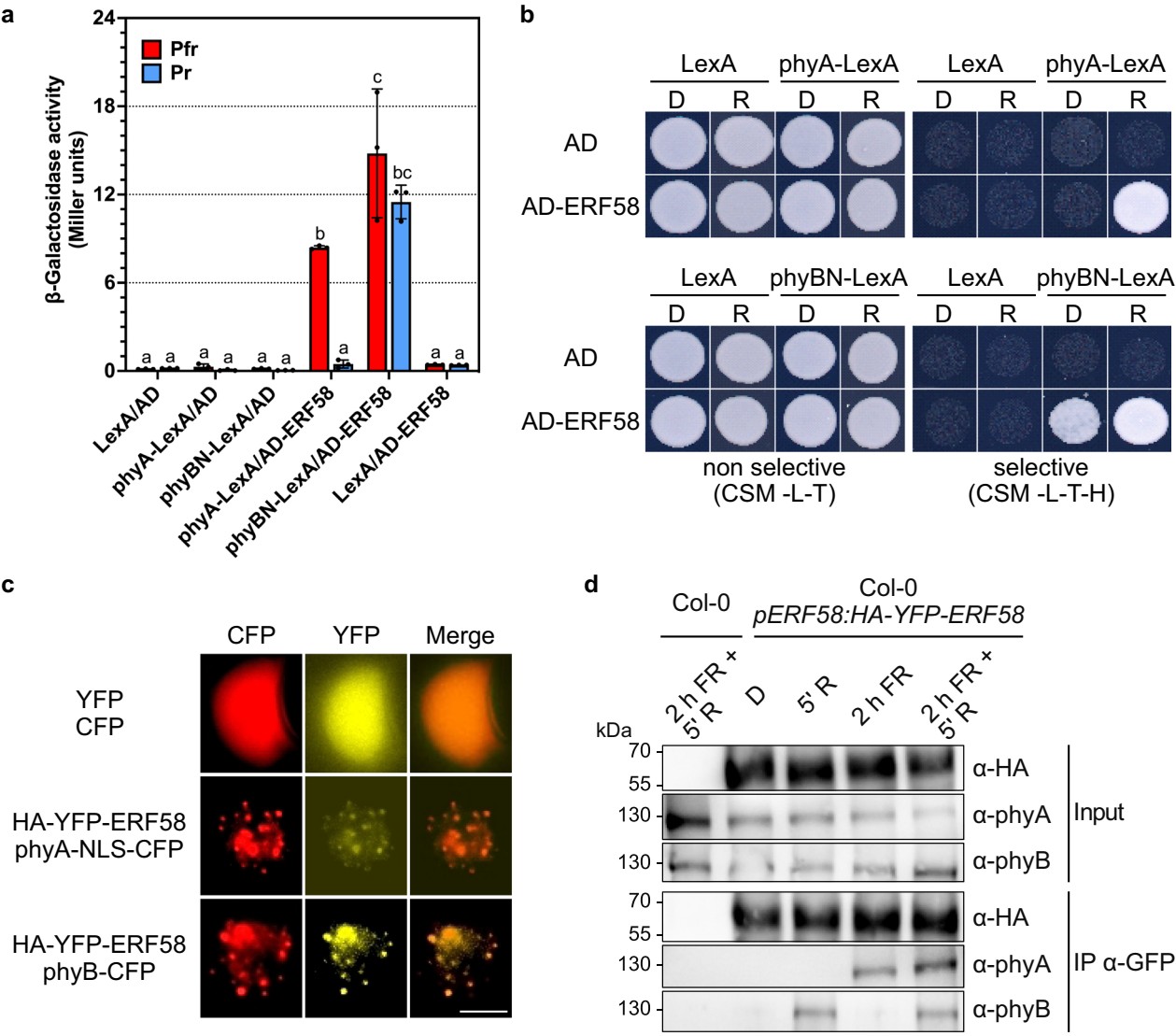

**Fig. 1 ERF58 interacts with phyA and phyB. a** Y2H protein-protein interaction assay. Full-length phyA or the N-terminal half of phyB (phyBN) fused to LexA and ERF58 fused to the GAL4 AD were expressed in yeast. Yeast cells were grown in chromophore (PCB)-supplemented medium, exposed to R or FR light for 5 min to convert phytochromes to Pfr or Pr, and incubated in the dark for another 4 h. β-Gal activity was then measured using an ONPG assay. Bars show mean β-Gal activity of three replicates ±SD. Different letters indicate significant differences as determined by two-way ANOVA followed by post-hoc Tukey's HSD test; $p < 0.05$. **b** Y2H growth assay. phyA- or phyBN-LexA and AD-ERF58 were expressed in yeast. Yeast cells were grown on CSM -L-T plates or CSM -L-T-H plates supplemented with PCB. Plates were incubated in the dark (D) or R light (R). **c** Co-localisation of ERF58 with phyA and phyB in tobacco. *p35S:HA-YFP-ERF58* and either *p35S:PHYA-NLS-CFP* or *p35S:PHYB-CFP* were transiently co-expressed in tobacco leaf epidermis cells by agro infiltration; empty *p35S:YFP* and *p35S:CFP* vectors were used as control. YFP and CFP signals were detected by epifluorescence microscopy. Scale bar represents 5 µm. **d** Co-immunoprecipitation of phyA and phyB with ERF58. Stable transgenic Arabidopsis lines expressing *pERF58:HA-YFP-ERF58* in Col-0 background were used for Co-IP; Col-0 was used as negative control. Three day old dark-grown seedlings were treated for 5 min with R light, for 2 h with FR light, for 2 h with FR light followed by 5 min R light, or kept in the dark and used for Co-IP. Eluate fractions were analysed by SDS-PAGE and immunoblotting with α-phyA, α-phyB, or α-HA antibodies. Experiments in **b–d** were repeated three times with similar results.

Under phyA-ON conditions, only phyA can induce the completion of germination of wild-type seeds, while under phyB-ON conditions, phyB is required for germination completion; under phyA- and phyB-OFF conditions, both phyA and phyB are inactive (Fig. 2a, b). Final cumulative germination percentages of *erf55-1*, *erf58-1*, and *erf58-2* seeds were increased under phyB-ON conditions compared to the wild-type and other *erf* mutants (Fig. 2c, d; Supplementary Fig. 3a, b). Likewise, under phyA-ON conditions, *erf55-1*, *erf58-1*, and *erf58-2* mutants had a higher percentage of germination completion than the wild-type (Fig. 2e, f; Supplementary Fig. 3c). In contrast, overexpression of

ERF58 reduced the completion of germination compared to the wild-type under both phyA- and phyB-ON conditions (Fig. 2c–f; Supplementary Fig. 3b, c). Complementation of *erf55-1* and *erf58-2* by expression of *pERF55:ERF55* and *pERF58:HA-YFP-ERF58* confirmed that defects in *ERF55* and *ERF58* are responsible for the increased final cumulative germination percentage of *erf55-1* and *erf58-2* (Supplementary Figs. 4 and 5). Unlike the *pif1* mutant[7], *erf55-1*, *erf58-1*, and *erf58-2* single mutants still required light for the completion of germination, but we observed that *erf55-1 erf58-2* double mutant seeds completed germination in the dark (Fig. 2c–f). Taken together, we conclude

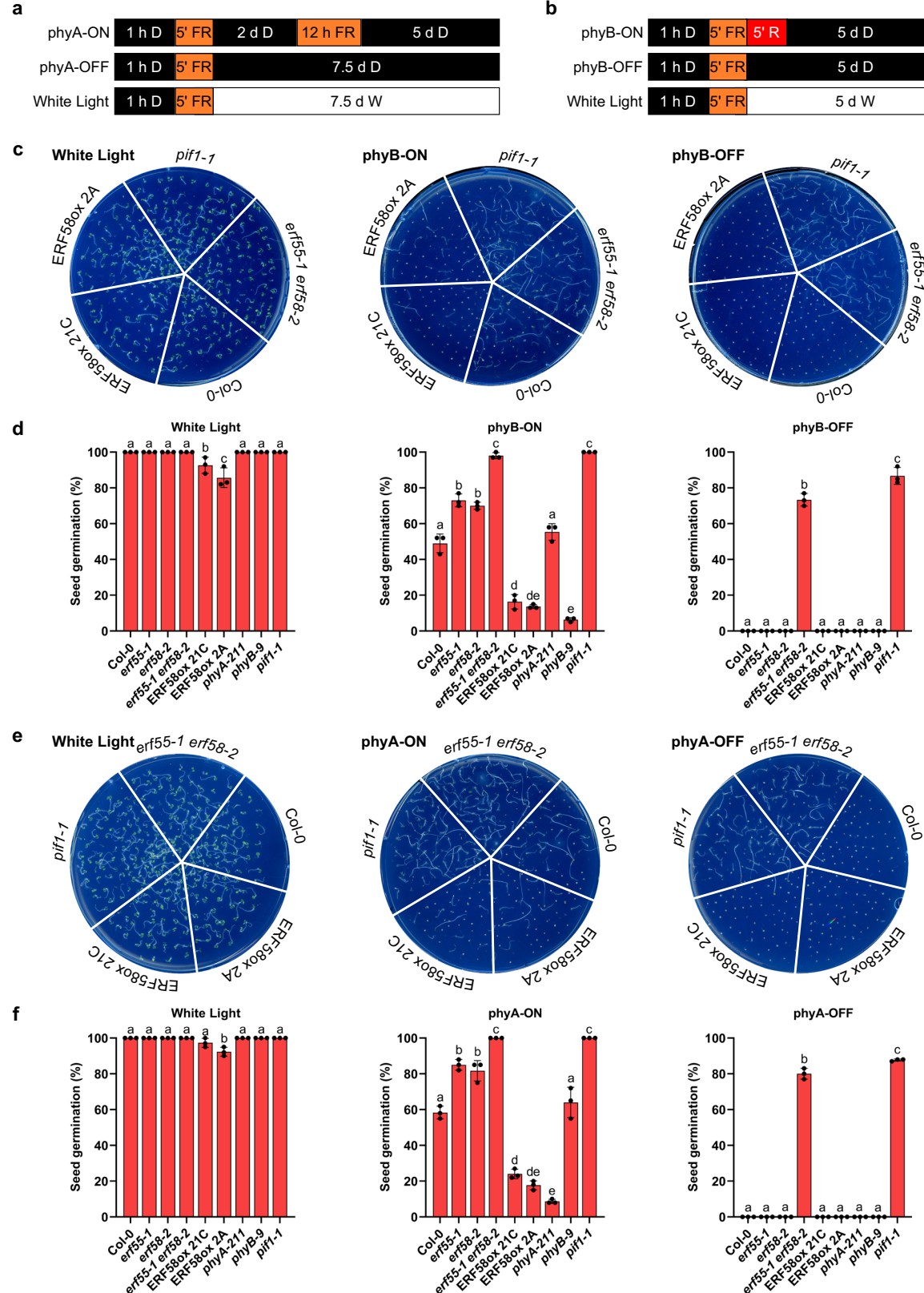

that ERF55 and ERF58 are negative regulators of the completion of seed germination required for suppression of germination completion in the dark.

**PhyA and phyB repress *ERF55* and *ERF58* expression.** Since ERF55 and ERF58 play a role in light-regulation of the

completion of seed germination, we used RT-qPCR to test if *ERF55* and *ERF58* transcript levels in germinating seeds are altered in response to phyA- or phyB-specific light treatments (Supplementary Fig. 6a, b). Under both phyA-ON and phyB-ON conditions, *ERF55* and *ERF58* transcript abundance was reduced compared to phyA-OFF and phyB-OFF, depending on phyA and

**Fig. 2 ERF55 and ERF58 inhibit the completion of seed germination. a, b** Light treatments used to test for phyA- and phyB-dependent completion of seed germination. Seeds were imbibed for 1 h in the dark (D), treated for 5 min with FR light and then exposed to light conditions that induce the completion of seed germination either through phyA (phyA-ON: 2 days in D → 12 h in FR light → 5 days in D) or phyB (phyB-ON: 5 min in R light → 5 days in D). phyA and phyB are inactive under phyA-OFF and phyB-OFF conditions, respectively. Germination completion in white (W) light was used to confirm viability of seeds. **c–f** phyA- and phyB-dependent completion of seed germination. Germination completion of wild-type (Col-0), different mutants, and two independent ERF58ox lines (21C and 2A) was tested under phyB-specific (**c, d**) or phyA-specific conditions (**e, f**). **c, e** Representative pictures of plates used for quantification of cumulative seed germination. **d, f** Bars show mean cumulative germination percentages of three replicates ±SD. Different letters indicate significant differences as determined by one-way ANOVA followed by post-hoc Tukey's HSD test; $p < 0.05$.

phyB, respectively (Supplementary Fig. 6c, d). ERF58 protein levels in ERF58ox lines were not or only slightly affected by light treatments (Supplementary Fig. 6e, f).

**ERF55 and ERF58 bind to the promoter of _PIF1_ and _SOM_.** PIF1 and SOM negatively regulate the completion of seed germination downstream of phytochromes[7,13]. We therefore quantified _PIF1_ and _SOM_ expression in wild-type and _erf55-1 erf58-2_ seeds exposed to phyA- or phyB-specific light treatments. _PIF1_ and _SOM_ transcript levels were reduced in _erf55-1 erf58-2_ compared to the wild-type, consistent with a role for ERF55 and ERF58 in suppressing the completion of germination (Fig. 3a, b).

To obtain further support for a function of ERF55 and ERF58 in promoting the expression of _PIF1_ and _SOM_, we used a transactivation assay in infiltrated tobacco leaves. HA-YFP-ERF55, HA-YFP-ERF58, or HA-YFP were co-expressed in tobacco leaves with _PIF1_ or _SOM_ promoter:luciferase reporter constructs. All luminescence signals were normalised to _p35S:GUS_ expression (Fig. 3c). The expression of ERF55 or ERF58 resulted in 2- to 4-fold increased activation of the promoter of _PIF1_ or _SOM_ compared to the empty HA-YFP vector control (Fig. 3d, e). PIF1 and ABI5 were used as positive controls as they have previously been shown to promote _PIF1_ and _SOM_ promoter activity[13,28].

Next, we used _PIF1_ and _SOM_ promoter fragments to identify regions of these promoters that are sufficient to enhance the expression of the luciferase reporter when ERF55 or ERF58 are co-transformed. Different _PIF1_ and _SOM_ promoter fragments resulted in activation of the luciferase reporter and therefore contain potential binding sites for ERF55 and ERF58 (Supplementary Figs. 7 and 8). We decided to use the −800…−1028 region of the _PIF1_ promoter and the −670…−840 region of the _SOM_ promoter for further experiments.

To investigate if ERF55 and ERF58 directly associate with the promoter of _PIF1_ and _SOM_, we expressed them as MBP fusion proteins in _E. coli_, purified them, and used them for EMSAs (Electrophoretic mobility shift assays) with biotin-labelled _PIF1_ or _SOM_ promoter fragments; an unlabelled fragment containing the DRE element to which ERF55 and ERF58 bind was used as competitor (Fig. 3f, g). Both ERF55 and ERF58 caused an upward shift of the _PIF1_ and _SOM_ promoter fragments, suggesting ERF55 and ERF58 bind to them in vitro.

We then performed ChIP-qPCR using _p35S:HA-YFP-ERF58_ (ERF58ox) and _p35S:YFP-HA_ transgenic lines. Chromatin was isolated from seeds germinating for 6 h and used for ChIP with α-GFP coupled magnetic beads followed by qPCR for quantification of _PIF1_ and _SOM_ promoter fragments in the eluate fraction. _PIF1_ and _SOM_ promoter fragments were strongly enriched in the ERF58ox sample compared to the YFP-HA negative control, suggesting that ERF58 associates with the _PIF1_ and _SOM_ promoter _in planta_ (Fig. 4a–d).

We also investigated the completion of seed germination of _erf55-1 erf58-2_ in _SOM_ and _PIF_ mutant or overexpression background (Supplementary Fig. 9). The set of seed batches used for the experiments in Supplementary Fig. 9 generally show a

lower final cumulative germination percentage than seed batches used in other experiments due to different conditions during seed propagation. In phyA- and phyB-OFF conditions, we observed a low final cumulative germination percentage for the _erf55-1 erf58-2_ mutant and no seeds that completed germination for _som-3_; in contrast, final cumulative germination was almost 100% for the _erf55-1 erf58-2 som-3_ triple mutant (Supplementary Fig. 9a). Thus, _ERF55/ERF58_ possibly control the completion of seed germination also through a second _SOM_-independent pathway. In contrast to _erf55-1 erf58-2_, _pif1-1_ and _erf55-1 erf58-2 pif1-1_ mutants completed germination even under phyA- and phyB-OFF conditions (Supplementary Fig. 9b). Potential explanations for this observation are that _PIF1_ expression is only partially repressed in _erf55-1 erf58-2_ or that _PIF1_ could affect the completion of seed germination also independently of _ERF55_ and _ERF58_. Consistent with a model in which ERF55 and ERF58 are acting upstream of _PIF1_ and _SOM_ to promote their transcription, we found that expression of _PIF1_ and _SOM_ under the control of the constitutively active CaMV _35S_ promoter can partially complement the _erf55-1 erf58-2_ mutant (Supplementary Fig. 9a, b).

Overall, we conclude that _PIF1_ and _SOM_ are direct targets of ERF58 and possibly ERF55 and that ERF55/ERF58 control the completion of seed germination through both PIF1/SOM-dependent and -independent mechanisms.

**Phytochromes inhibit the DNA-binding activity of ERF55/ERF58.** In seeds germinating under phyB-ON conditions, transcript levels of _ERF55_ and _ERF58_ are only reduced by about 50% when compared to seeds given a phyB-OFF treatment, and ERF58 protein stability is not or only weakly affected by light (Supplementary Fig. 6). Thus, we hypothesised that light might regulate ERF55/ERF58 also at other levels. Association with target promoters is another layer at which transcription factors can be controlled and therefore we used ChIP-qPCR to test if light has an effect on binding of ERF58 to the promoter of _PIF1_ or _SOM_. For this experiment, we slightly adapted the light treatment of seeds. Only 20 min before harvesting the samples for ChIP, a light treatment was given to activate phyB (phyB-OFF-ON) in seeds previously exposed to phyB-OFF conditions or to inactivate phyB (phyB-ON-OFF) in seeds previously exposed to a phyB-ON treatment (Fig. 4a). The short time between the light treatment setting the final Pfr/Ptot level of phyB and harvesting the seeds for ChIP reduces indirect effects of light on the amount of ERF58 binding to target promoters and restricts the analysis to the influence of phyB because phyA is not yet present[29]; immunoblot analysis furthermore confirmed that ERF58 protein levels in phyB-OFF and phyB-OFF-ON as well as in phyB-ON and phyB-ON-OFF conditions are very similar; in contrast, ERF58 protein levels are slightly higher in phyB-OFF/phyB-OFF-ON than phyB-ON/phyB-ON-OFF (Fig. 4b).

ChIP-qPCR for seeds exposed to phyB-ON vs. phyB-ON-OFF showed that association of ERF58 with the _PIF1_ and _SOM_ promoter is strongly enhanced by the 10 min FR pulse in the phyB-ON-OFF treatment, suggesting that inactivation of phyB

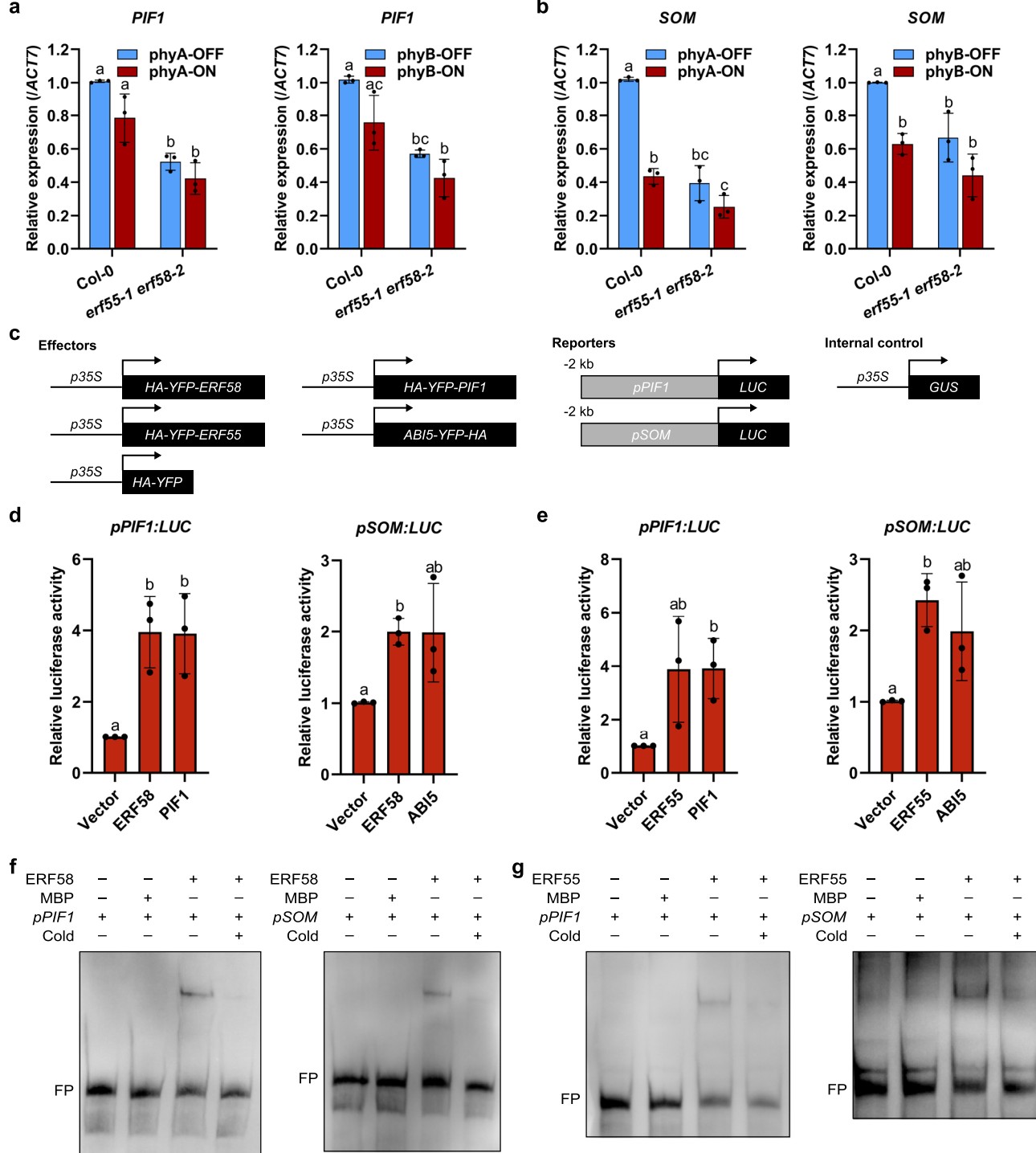

**Fig. 3 ERF58 directly binds to the promoter of *PIF1* and *SOM* to induce their expression. a**, **b** RT-qPCR analysis of *PIF1* (**a**) and *SOM* (**b**) expression in Col-0 and *erf55-1 erf58-2* seeds germinating under phyA-ON/OFF or phyB-ON/OFF conditions. Light conditions are described in Supplementary Fig. 6a, b. *ACT7* was used as an internal control. Values are means of three replicates ±SD. Different letters indicate significant differences as determined by two-way ANOVA followed by post-hoc Tukey's HSD test; *p* < 0.05. **c** Constructs used for transactivation assays. **d**, **e** Transactivation assay in tobacco. Combinations of effector (*p35S:HA-YFP-ERF58* or *p35S:HA-YFP-ERF55*; *p35S:HA-YFP* was used as negative control; *p35S:HA-YFP-PIF1* and *p35S:ABI5-YFP-HA* were used as positive controls) and reporter constructs (*pPIF1 −1...−2000:LUC* or *pSOM −1...−2000:LUC*) and *p35S:GUS* (used for normalisation) were transiently co-expressed in tobacco leaf epidermis cells by agro infiltration. LUC and GUS activity was measured in protein extracts from infiltrated leaves. Bars show mean relative LUC activity (LUC activity divided by GUS activity) of three replicates ±SD. Different letters indicate significant differences as determined by *t*-test and Holm–Bonferroni method; *p* < 0.05. **f**, **g** EMSAs. Biotin-labelled *PIF1* and *SOM* promoter fragments (*pPIF1 −801...−1031* and *pSOM −621...−840*) were incubated with MBP-ERF58, MBP-ERF55, or MBP alone (negative control). Samples were analysed by native PAGE. Gels were blotted onto nylon membranes and signals were detected by streptavidin-coupled horseradish peroxidase and ECL. FP, free probe; Cold, excess of unlabelled DNA fragment containing the DRE motif to which ERF55/ERF58 bind. Experiments were repeated at least three times with similar results.

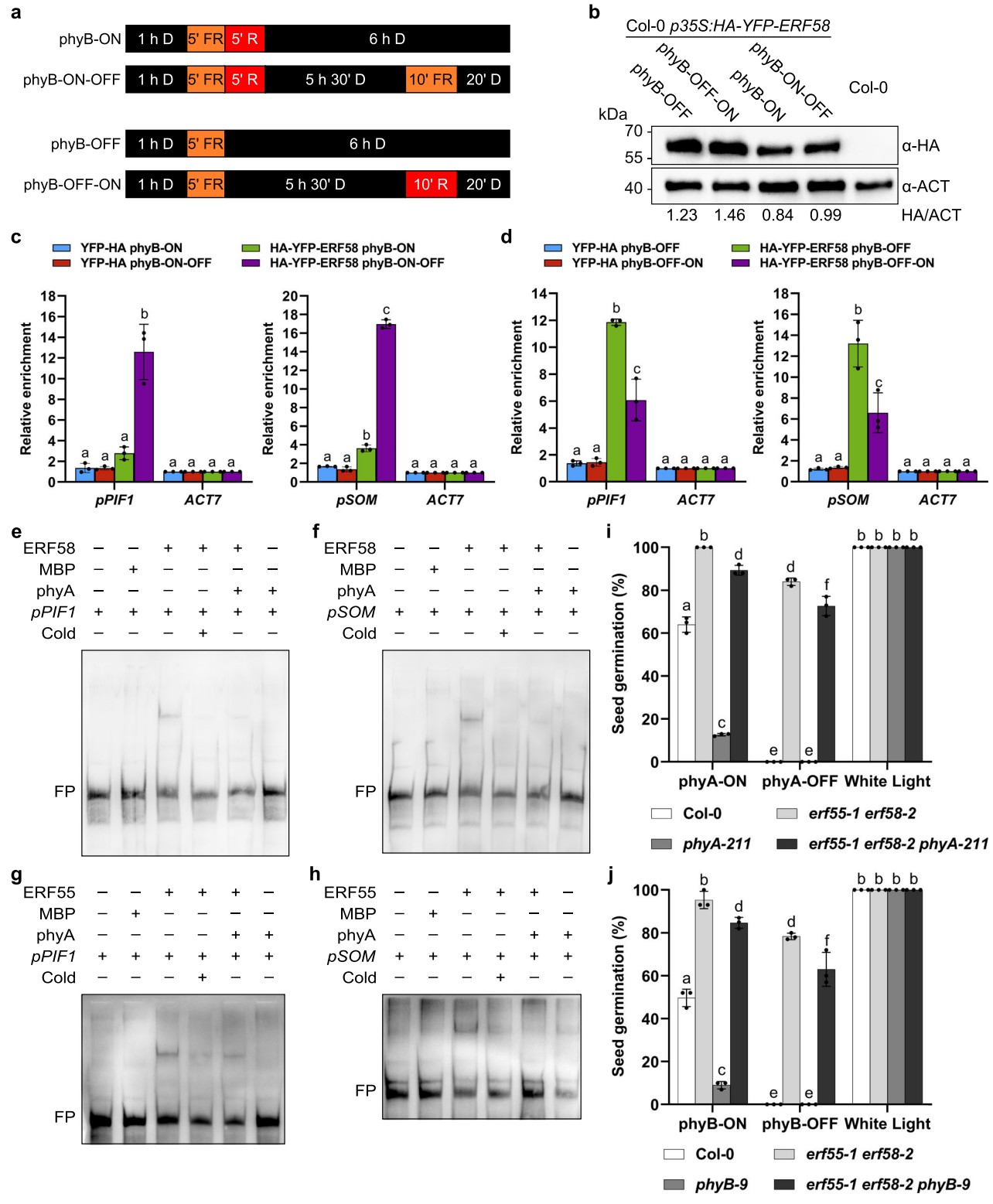

promotes binding of ERF58 to target promoters (Fig. 4c). In contrast, when comparing phyB-OFF vs. phyB-OFF-ON, the R pulse applied under phyB-OFF-ON conditions reduced binding of ERF58 to the promoter of *PIF1* and *SOM*, indicating that activation of phyB promotes dissociation of ERF58 from target promoters (Fig. 4d). Thus, association of ERF58 with promoters of target genes is regulated in a R/FR reversible manner, suggesting it is controlled by phytochromes. Since phyA is not present in seeds incubated on water for only few hours[29],

light-regulation of ERF58 promoter association at this early time point is likely primarily conferred by phyB, while at later time points also phyA could play a role.

The rapid regulation of ERF58 promoter dissociation/association in response to R and FR pulses indicates this could be a direct effect of phytochromes binding to ERF58. We therefore used ERF58 truncations for Y2H assays to identify the domain of ERF58 interacting with phytochromes (Supplementary Fig. 10). We found that both the C-terminal domain and the AP2 domain

**Fig. 4 Phytochromes inhibit the DNA-binding activity of ERF55 and ERF58. a** Light conditions used in the ChIP experiment. Seeds were imbibed for 1 h in the dark (D) and treated for 5 min with FR light to inactivate phyB. Then, phyB was kept in the inactive state by incubation in the dark (phyB-OFF) or activated by 5 min R light before returning the seeds into darkness (phyB-ON). Twenty min before harvesting for ChIP, seeds in phyB-OFF conditions were exposed to R light for 10 min to activate phyB (phyB-OFF-ON), and seeds in phyB-ON conditions were treated with FR light for 10 min to inactivate phyB (phyB-ON-OFF). **b** HA-YFP-ERF58 protein levels in ERF58ox seeds. Col-0 *p35S:HA-YFP-ERF58* (line 21C) seeds were treated as described in **a**. Total protein was then extracted and analysed by SDS-PAGE and immunoblotting with α-HA; α-ACT was used to detect ACTIN as loading control. Col-0 was included as negative control. Signals detected by α-HA and α-ACT were quantified using ImageJ; numbers below the membrane show the HA/ACT ratio. **c, d** ChIP-qPCR. Seeds of Col-0 *p35S:HA-YFP-ERF58* (line 21C) and Col-0 *p35S:YFP-HA* (negative control) were incubated and subject to light treatments as described in **a**. Chromatin was isolated and HA-YFP-ERF58 bound DNA fragments were purified using α-GFP coupled magnetic beads. *PIF1* and *SOM* promoter fragments in the input and eluate fractions were detected using qPCR and specific primers. *ACT7* was used for normalisation. Values show enrichment of *PIF1* and *SOM* promoter fragments in the eluate fraction compared to the input fraction and are means of three replicates ±SD. **e–h** EMSAs. Biotin-labelled *pPIF1* −801...−1031 (**e, g**) or *pSOM* −621...−840 (**f, h**) promoter fragments were incubated with MBP-ERF58 (**e, f**), MBP-ERF55 (**g, h**), or MBP alone (negative control), and light-activated phyA. Samples were analysed by native PAGE. Gels were blotted onto nylon membranes and signals were detected by streptavidin-coupled horseradish peroxidase and ECL. FP, free probe; Cold, excess of unlabelled DNA fragment containing the DRE motif to which ERF55/ERF58 bind. Zinc blot analysis of EMSA samples is shown in Supplementary Fig. 11b, c. **i, j** Test for completion of germination. The completion of germination of indicated genotypes was tested under phyA-ON/OFF (**i**) and phyB-ON/OFF conditions (**j**). Bars show mean cumulative germination percentages of three replicates ±SD. **c, d, i, j** Different letters indicate significant differences as determined by two-way ANOVA followed by post-hoc Tukey's HSD test. **c, d** $p < 0.01$. **i, j** $p < 0.05$. Experiments in **b** and **e–h** were repeated at least three times with similar results.

of ERF58 bind phyA and phyB. The AP2 domain is highly conserved in AP2/ERF transcription factors and required for binding to DNA. We therefore hypothesised that interaction of light-activated phytochromes with ERF55 and ERF58 could block their DNA-binding site, thereby preventing binding of free ERF55/ERF58 to target promoters and displacing promoter-bound ERF55/ERF58 from their cognate motifs. To test such a direct effect of phytochromes on the DNA-binding capacity of ERF55 and ERF58, we expressed photoactive full-length phyA in an *E. coli* strain engineered to synthesise phytochromobilin, the natural chromophore of seed plant phytochromes (Supplementary Fig. 11a). Purified phyA was then used for EMSAs with ERF55/ERF58 and biotin-labelled *PIF1* and *SOM* promoter fragments (Fig. 4e–h; Supplementary Fig. 11b, c). ERF55 and ERF58 caused an upward shift of *PIF1* and *SOM* promoter fragments, which is largely abolished when phyA is supplemented. Expression of phyB in *E. coli* was too low to obtain sufficient amounts for EMSAs. Overall, ChIP-qPCR and EMSA data are consistent with a mechanistic model in which light-activated phyA and phyB can bind ERF55/ERF58 to prevent their association with target promoters.

To genetically test if ERF55 and ERF58 act downstream of phyA and phyB, we generated *erf55-1 erf58-2 phyA-211* and *erf55-1 erf58-2 phyB-9* triple mutants. *erf55-1 erf58-2* is almost fully epistatic over *phyA-211* and *phyB-9* and we observed a high final cumulative germination percentage for the two triple mutants in phyA- and phyB-OFF conditions (Figs. 2a, b and 4i, j). This finding is consistent with a model in which the repression of ERF55 and ERF58 activity by light-activated phytochromes promotes the completion of seed germination.

**ERF55/ERF58 regulate genes encoding ABA metabolic enzymes.** The phytohormone ABA is an important inhibitor of the completion of seed germination[2]. We observed that the inhibitory effect of ABA on germination completion is reduced in the *erf55-1 erf58-2* mutant compared to the wild-type, while it is increased in two independent ERF58ox lines (Fig. 5a), indicating that ABA could be involved in ERF55/ERF58-mediated repression of the completion of seed germination. Using RT-qPCR, we therefore quantified transcript levels of genes encoding ABA anabolic (*ABA1*, *ABA2*, *NCED6*, *NCED9*, *AAO3*) or catabolic (*CYP707A2*) enzymes[13,14]. Expression of genes that encode ABA anabolic enzymes was reduced in germinating *erf55-1 erf58-2* seeds compared to the wild-type, whereas expression of *CYP707A2*, a gene encoding an ABA catabolic enzyme, was

increased (Fig. 5b; Supplementary Fig. 12). For most genes coding for ABA metabolic enzymes that we tested, expression levels were similar in the wild-type and the *erf55-1 erf58-2* double mutant under phyA/phyB-ON conditions, while the difference was much stronger under phyA/phyB-OFF conditions. This is consistent with a model, in which light-activated phytochromes repress ERF55/ERF58 action, and suggests a critical function of ERF55 and ERF58 in light regulation of genes encoding ABA metabolic enzymes. In line with ERF55/ERF58-mediated activation of genes encoding ABA anabolic enzymes and repression of genes encoding ABA catabolic enzymes, the level of endogenous ABA in germinating seeds is lower in *erf55-1 erf58-2* than in the wildtype; conversely, ABA levels are increased in ERF58ox seeds compared to the wild-type (Fig. 5c).

Previous studies suggest that *ABA2*, a single copy gene encoding an ABA anabolic enzyme, is not regulated by PIF1 and SOM[12,13]. However, we identified a DRE element in the *ABA2* promoter, which is known as binding site for different members of the AP2/ERF family (Fig. 5d, e). In EMSAs, ERF55 and ERF58 bound to the *ABA2* promoter fragment containing the DRE element, while mutating the DRE element abolished promoter association (Fig. 5e, f; Supplementary Fig. 13a). Furthermore, using EMSA, we also observed association of ERF55 and ERF58 with *NCED9* and *AAO3* promoter fragments, which was strongly reduced when light-activated phyA was added to the assay (Supplementary Fig. 14).

To confirm association of ERF58 with the *ABA2* promoter *in planta*, we performed ChIP-qPCR. Similar to association with the *PIF1* and *SOM* promoter, binding of ERF58 to the *ABA2* promoter was regulated in a R/FR reversible manner, with FR and R light promoting binding and unbinding, respectively (Figs. 4a and 5g). Taken together, these results suggest that ERF55 and ERF58 inhibit the completion of seed germination by regulating genes encoding ABA metabolic enzymes and increasing ABA levels.

**ERF55 and ERF58 promote ABA downstream signalling.** In addition to genes coding for ABA metabolic enzymes, genes that have a function in ABA downstream signalling are also targets potentially regulating the completion of seed germination[3]. ABI5 is a key factor of ABA signal transduction[30] and we found that *ABI5* transcript levels in germinating seeds are lower in *erf55-1 erf58-2* than the wild-type (Fig. 6a). Like the *ABA2* promoter, the promoter of *ABI5* also contains a DRE element (Fig. 6b, c). In EMSAs, both ERF55 and ERF58 bound a wild-type *ABI5*

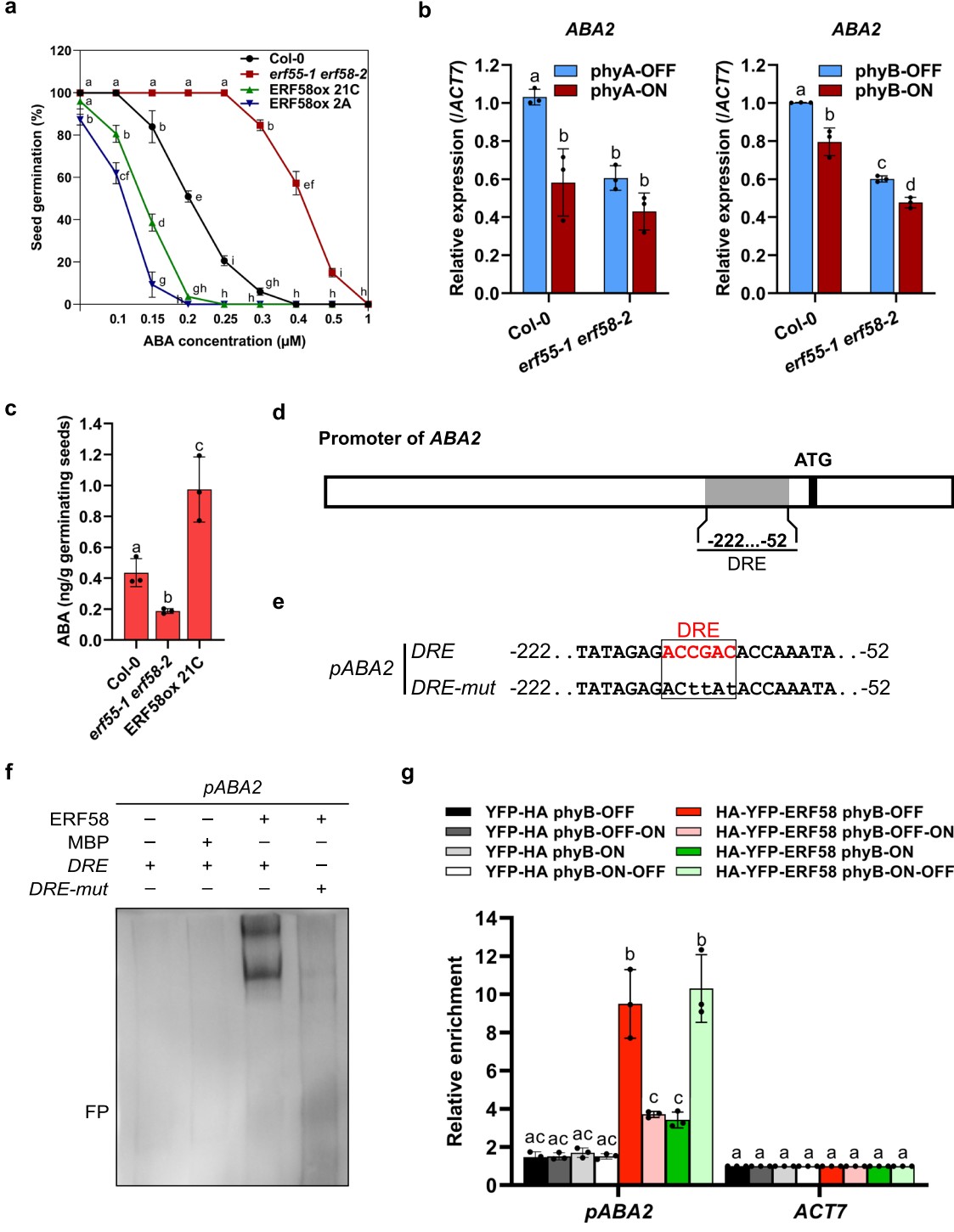

promoter fragment, while inserting point mutations into the DRE element abolished binding (Fig. 6c, d; Supplementary Fig. 13b). Finally, we used ChIP-qPCR to confirm association of ERF58 with the *ABI5* promoter in germinating seeds. We found that binding of ERF58 to the *ABI5* promoter is reversibly regulated by R and FR light, with a R light pulse inhibiting binding and a FR light pulse promoting it (Fig. 6e). Collectively, these findings suggest that ERF55 and ERF58 enhance the expression of *ABI5* and thereby promote ABA downstream signalling.

In agreement with a model in which *ERF55* and *ERF58* are acting upstream of *ABI5* to promote its transcription, we found that expression of *ABI5* under the control of the constitutively

active CaMV *35S* promoter partially complements the *erf55-1 erf58-2* mutant phenotype (Supplementary Fig. 9c). In sets of seed batches with a low final cumulative germination percentage, few *erf55-1 erf58-2* double mutant seeds completed germination in phyA- and phyB-OFF conditions, while *abi5-8* did not complete germination at all under these conditions. In contrast, the final cumulative germination percentage for the *erf55-1 erf58-2 abi5-8* triple mutant in phyA- or phyB-OFF conditions was much higher compared to the *erf55-1 erf58-2* and *abi5-8* parental lines, suggesting that ERF55 and ERF58 repress the completion of seed germination also partially independent of *ABI5* (Supplementary Fig. 9c).

**Fig. 5 ERF55 and ERF58 regulate the expression of genes encoding ABA metabolic enzymes and increase ABA levels. a** Completion of germination of wild-type (Col-0), *erf55-1 erf58-2* double mutant, and ERF58ox (two independent lines) seeds on ddH$_2$O and increasing concentrations of ABA. Seeds were exposed to W light for 6 h, and then incubated in the dark at 22 °C for 7 days before scoring for completion of germination. Data points show mean cumulative germination percentages of three replicates ±SD. **b** RT-qPCR analysis of *ABA2* expression levels in Col-0 and *erf55-1 erf58-2* seeds germinating under phyA-ON/OFF or phyB-ON/OFF conditions; data for other genes encoding ABA metabolic enzymes are shown in Supplementary Fig. 12. Light conditions are described in Supplementary Fig. 6a, b. *ACT7* was used as an internal control. Values are means of three replicates ±SD. **c** Quantification of endogenous ABA levels in Col-0, *erf55-1 erf58-2*, and ERF58ox seeds after 24 h germination in the dark. Values are means of three replicates ±SD. Letters indicate levels of significance as determined by *t*-test and Holm–Bonferroni method; *p* < 0.05. **d**, **e** Schematic representation of the *ABA2* promoter. The promoter fragment used for EMSA is shown in **e**. **f** EMSA. Biotin-labelled *ABA2* −52...−222 promoter fragments containing a wild-type or mutated DRE motif were incubated with MBP-ERF58 or MBP alone (negative control). Samples were analysed by native PAGE. The gel was blotted onto a nylon membrane and signals were detected by streptavidin-coupled horseradish peroxidase and ECL. FP, free probe. The experiment was repeated six times with similar results. Data for ERF55 are shown in Supplementary Fig. 13a. **g** ChIP-qPCR. Seeds of Col-0 *p35S:HA-YFP-ERF58* (line 21C) and Col-0 *p35S:YFP-HA* (negative control) were germinated and subject to light treatments as described in Fig. 4a. Chromatin was isolated and HA-YFP-ERF58 bound DNA fragments were purified using α-GFP coupled magnetic beads. qPCR and specific primers were used to detect the *ABA2* −52...−222 promoter fragment in the eluate fraction. *ACT7* was used for normalisation. Values show enrichment of the *ABA2* promoter fragment in the eluate fraction compared to the input fraction and are means of three replicates ±SD. **a**, **b**, **g** Different letters indicate significant differences as determined by two-way ANOVA followed by post-hoc Tukey's HSD test. **a**, **b** *p* < 0.05. **g** *p* < 0.01.

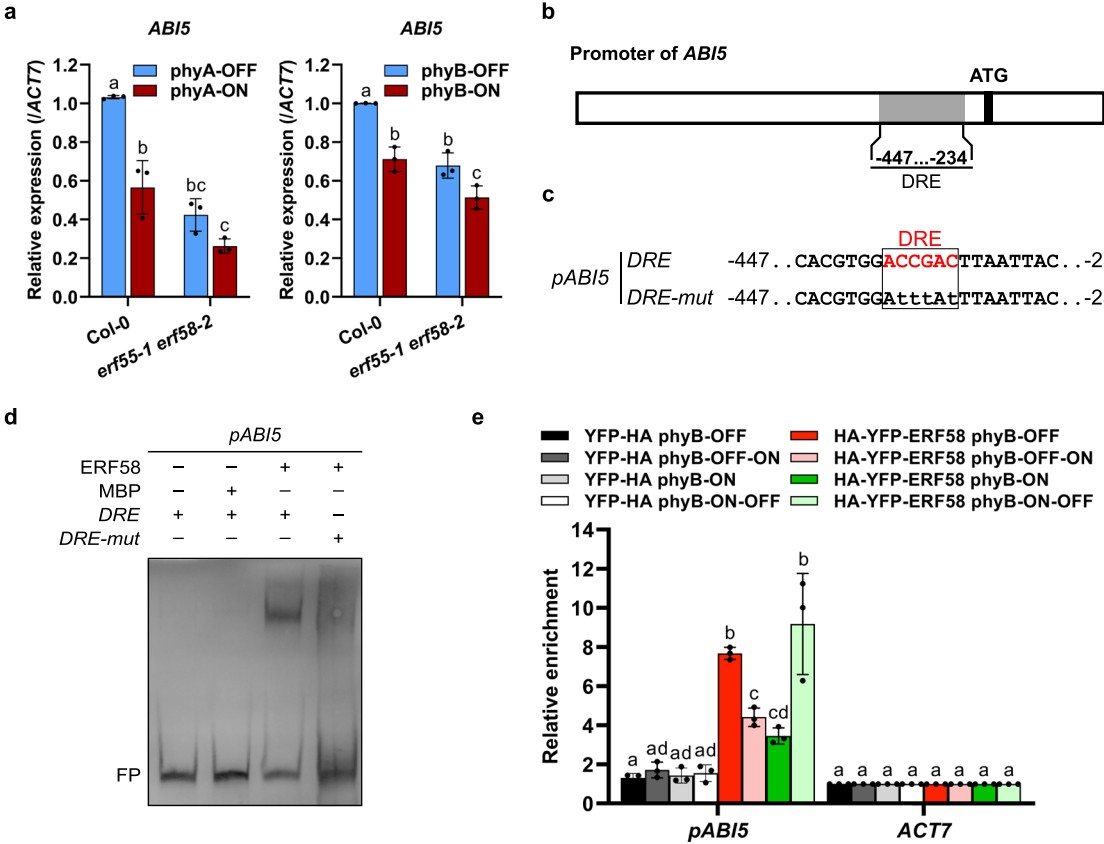

**Fig. 6 ERF55 and ERF58 upregulate *ABI5* expression. a** RT-qPCR analysis of *ABI5* expression levels in Col-0 and *erf55-1 erf58-2* seeds germinating under phyA-ON/OFF or phyB-ON/OFF conditions. Light conditions are described in Supplementary Fig. 6a, b. *ACT7* was used as an internal control. Values show means of three replicates ±SD. **b**, **c** Schematic representation of the *ABI5* promoter. The promoter fragment used for EMSA is shown in **c**. **d** EMSA. Biotin-labelled *ABI5* −234...−447 promoter fragments containing a wild-type or mutated DRE motif were incubated with MBP-ERF58 or MBP alone (negative control). Samples were analysed by native PAGE. The gel was blotted onto a nylon membrane and signals were detected by streptavidin-coupled horseradish peroxidase and ECL. FP, free probe. The experiment was repeated five times with similar results. Data for ERF55 are shown in Supplementary Fig. 13b. **e** ChIP-qPCR. Germinating seeds of Col-0 *p35S:HA-YFP-ERF58* (line 21C) and Col-0 *p35S:YFP-HA* (negative control) were subject to light treatments as described in Fig. 4a. Chromatin was isolated and HA-YFP-ERF58 bound DNA fragments were purified using α-GFP coupled magnetic beads. qPCR and specific primers were used to detect the *ABI5* −234...−447 promoter fragment in the eluate fraction. *ACT7* was used for normalisation. Values show enrichment of the *ABI5* promoter fragment in the eluate fraction compared to the input fraction and are means of three replicates ±SD. **a**, **e** Different letters indicate significant differences as determined by two-way ANOVA followed by post-hoc Tukey's HSD test. **a** *p* < 0.05. **e** *p* < 0.01.

**ERF55/ERF58 regulate genes encoding GA metabolic enzymes**. Light promotes the completion of germination in part by increasing the levels of bioactive gibberellins, which act antagonistically to ABA[2,13]. Therefore, we tested completion of germination of wild-type, *erf55-1 erf58-2*, and ERF58ox seeds on medium containing different concentrations of the bioactive gibberellin GA3 or the gibberellin biosynthesis inhibitor paclobutrazol (PAC). We observed that the *erf55-1 erf58-2* double mutant is hypersensitive to GA3 compared to the wild-type, while two independent ERF58ox lines are hyposensitive (Fig. 7a). In addition, wild-type and ERF58ox lines did not complete germination on medium containing PAC, whereas PAC inhibited the completion of germination of *erf55-1 erf58-2* only at higher concentrations (Fig. 7b). Therefore, ERF55 and ERF58 could regulate the completion of seed germination partially through GA.

Using RT-qPCR, we then quantified transcript levels of genes encoding GA metabolic enzymes in germinating seeds[15]. Expression of genes that encode GA anabolic enzymes (*GA3ox1*, *GA3ox2*) was upregulated in *erf55-1 erf58-2* compared to the wild-type, while expression of genes encoding GA catabolic enzymes (*GA2ox2*, *GA2ox4*) was reduced (Fig. 7c; Supplementary Fig. 15). Similar to genes that encode ABA metabolic enzymes, expression levels of most genes encoding GA metabolic enzymes that we tested differed more strongly between the wild-type and the *erf55-1 erf58-2* double mutant under phyA/phyB-OFF conditions than under phyA/phyB-ON conditions. This supports a function of ERF55 and ERF58 in light-regulation of the expression of these genes.

Like *ABA2*, *GA2ox4* is likely not a target of PIF1 and SOM but it is misregulated in *erf55-1 erf58-2* and its promoter contains a DRE element (Fig. 7d, e)[12]. EMSAs with biotin-labelled wild-type and DRE mutant *GA2ox4* promoter fragments showed that ERF55 and ERF58 associate with the *GA2ox4* promoter depending on an intact DRE element (Fig. 7e, f; Supplementary Fig. 13c).

Finally, we used ChIP-qPCR to show that ERF58 associates with the *GA2ox4* promoter in germinating seeds. Similar to other promoters bound by ERF58, also binding to the *GA2ox4* promoter was enhanced when phyB was in the inactive state, i.e. under phyB-OFF or phyB-ON-OFF conditions, while binding was lower under phyB-ON or phyB-OFF-ON conditions, where Pfr levels are high and phyB binds to ERF58 (Fig. 7g).

Overall we conclude that ERF55 and ERF58 affect the completion of seed germination also by regulation of genes encoding GA metabolic enzymes.

**ABA and GA regulate the expression of *ERF55* and *ERF58***. Feedback regulation is not uncommon in signalling pathways. To test for potential feedback regulation of *ERF55* and *ERF58* by ABA or GA, we quantified expression of *ERF55* and *ERF58* in seeds germinating on medium containing ABA or GA3. Compared to mock treated seeds, *ERF55* and *ERF58* transcript levels were reduced in seeds germinating in the presence of GA3, while ABA enhanced the expression of *ERF55* and *ERF58* (Fig. 8a–d).

Interestingly, we found that ERF58 protein levels in germinating ERF58ox seeds increased in response to ABA. The ERF58ox line expresses HA-YFP-ERF58 under the control of the constitutively active CaMV *35S* promoter, which is not regulated by ABA. Therefore, in addition to promoting *ERF58* expression, ABA also increases ERF58 protein stability (Fig. 8e). In contrast, GA3 did not have an obvious effect on ERF58 protein stability (Fig. 8f).

Taken together, ABA upregulates *ERF55* and *ERF58* transcript levels and ERF58 protein stability, while GA downregulates *ERF55* and *ERF58* expression, suggesting ERF55 and ERF58 are under positive feedback regulation by ABA and GA.

## Discussion

Seeds are generally much more tolerant to unfavourable environmental conditions than seedlings[2]. Therefore, the completion of seed germination – a critical point of no return in the life cycle of plants – must be reliably controlled. Here, we have shown that the AP2/ERF transcription factors ERF55 and ERF58 control the completion of seed germination downstream of phytochromes at different levels. First, they regulate *PIF1* and *SOM*, which are key transcriptional regulators of the completion of seed germination[7,13]; second, they control the expression of genes encoding ABA or GA metabolic enzymes; and third, they promote ABA downstream signalling. Action of ERF55 and ERF58 at all three levels can contribute to inhibition of the completion of germination.

ERF58 and ERF55 belong to a group of AP2/ERFs not containing a repression domain and therefore possibly exclusively enhancing the expression of direct target genes[16]. In line with this, all genes that we identified as direct targets of ERF58 and likely ERF55 are downregulated in the *erf55-1 erf58-2* mutant compared to the wild-type and all these genes repress the completion of germination. We also identified genes upregulated in *erf55-1 erf58-2* that promote germination completion, but these genes might be indirectly regulated by ERF55 and ERF58 (Fig. 9a).

The completion of germination of *erf55-1 erf58-2* seeds is not fully suppressed in the dark, i.e. under phyA- and phyB-OFF conditions. Therefore we propose a model in which light perceived by phyA and phyB downregulates ERF55 and ERF58 action. First, *ERF55* and *ERF58* transcript levels are reduced under phyA/phyB-ON conditions depending on phyA and phyB (Fig. 9a); in contrast, ERF58 protein stability appears not or only slightly affected by light. Second, using seeds germinating under phyB-ON conditions and exposed to a short FR light pulse, we have shown that converting phyB to the inactive Pr state leads to rapid association of ERF58 with target promoters. Conversely, a short R light pulse applied to seeds germinating under phyB-OFF conditions converts phyB to the active Pfr state and promotes the dissociation of ERF58 from target promoters. The R/FR reversible regulation of ERF58 target promoter association indicates that this effect depends on phyB. PhyA is not present in seeds imbibed for only a few hours[29], and it is likely that primarily phyB controls ERF58 target promoter association under the conditions used in the ChIP-qPCR experiments. However, EMSAs suggest that also phyA can regulate ERF58 and ERF55 target promoter binding, so phyA could use the same mechanism as phyB to control the association of ERF58 and possibly ERF55 with target promoters *in planta*. Phytochromes thus not only regulate the transcription of *ERF55* and *ERF58*, but also control the binding of ERF55 and ERF58 to the promoter of target genes (Fig. 9a)[29].

Light-activated phyA and phyB bind to the AP2-domain of ERF58 and thereby possibly directly block association of ERF58 with target promoters under phyA- and phyB-ON conditions (i.e. in light). Binding to phyA and phyB may be further stabilised by the interaction of phytochromes with the C-terminus of ERF58. Thus, in contrast to PIFs, which use dedicated motifs for binding phyA and phyB[31], ERF58 can bind phyA and phyB through its DNA-binding domain. The AP2-domain is the hallmark of AP2/ERF transcription factors[16] and binding to this domain could explain why most members of the A6 group of AP2/ERFs interact with phyA and phyB, and may also explain why CBF1, an AP2/ERF transcription factor from group A1, binds phyA and phyB[32]. In addition, phytochromes might also bind members of other

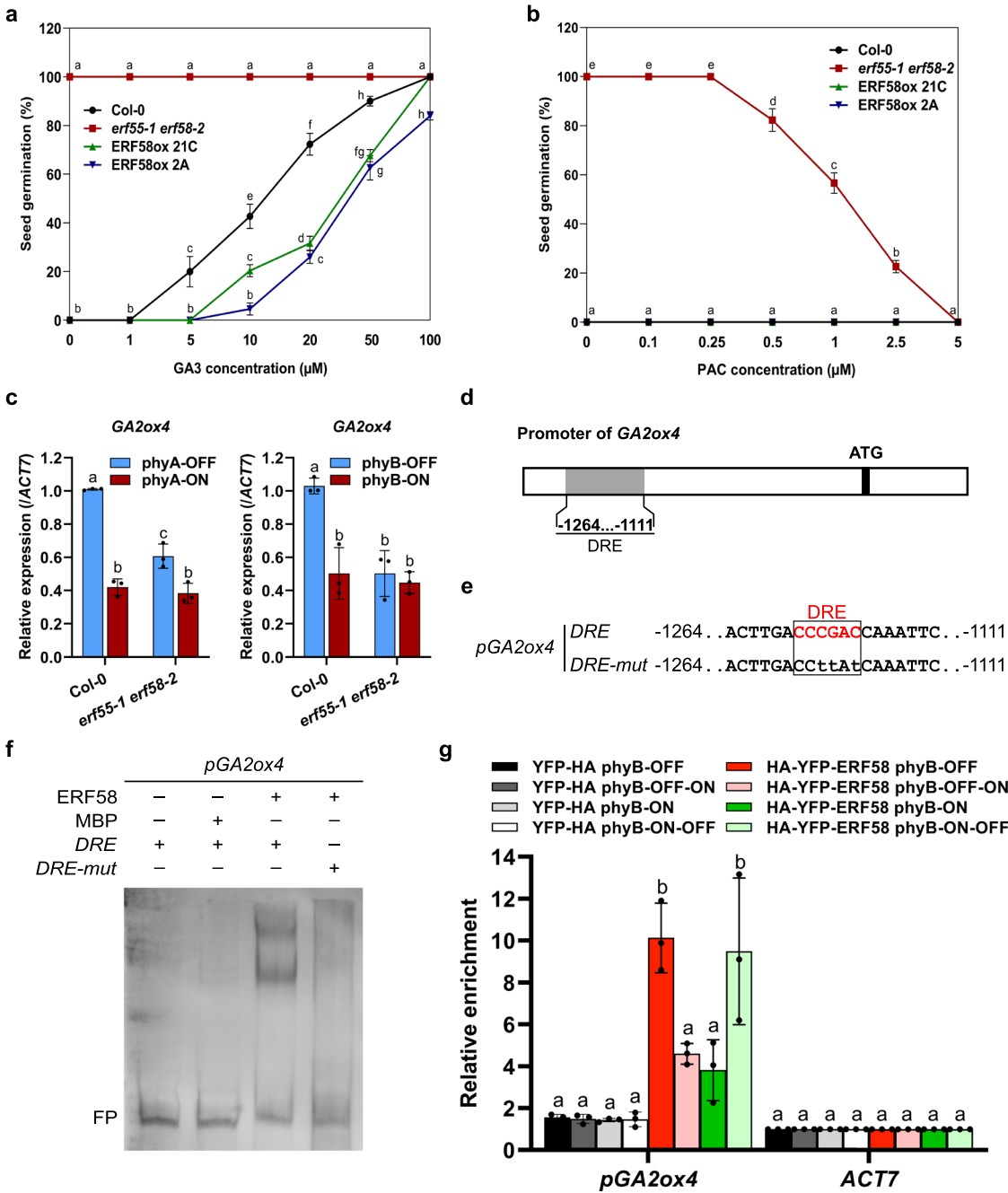

**Fig. 7 ERF55 and ERF58 regulate the expression of genes encoding GA metabolic enzymes. a, b** Completion of germination of wild-type (Col-0), *erf55-1 erf58-2* double mutant, and ERF58ox (two independent lines) seeds on ddH$_2$O and increasing concentrations of GA3 (**a**) or the GA biosynthesis inhibitor PAC (**b**). Seeds were not exposed to light (**a**) or exposed to W light for 6 h (**b**); after 7 days incubation in the dark at 22 °C, seeds were scored for completion of germination. Data points show mean cumulative germination percentages of three replicates ±SD. **c** RT-qPCR analysis of *GA2ox4* expression levels in Col-0 and *erf55-1 erf58-2* seeds incubated under phyA-ON/OFF or phyB-ON/OFF conditions; data for other genes encoding GA metabolic enzymes are shown in Supplementary Fig. 15. Light conditions are described in Supplementary Fig. 6a, b. *ACT7* was used as an internal control. Values show means of three replicates ±SD. **d, e** Schematic representation of the *GA2ox4* promoter. The promoter fragment used for EMSA is shown in **e**. **f** EMSA. Biotin-labelled *GA2ox4* −1111...−1264 promoter fragments containing a wild-type or mutated DRE motif were incubated with MBP-ERF58 or MBP alone (negative control). Samples were analysed by native PAGE. The gel was blotted onto a nylon membrane and signals were detected by streptavidin-coupled horseradish peroxidase and ECL. FP, free probe. The experiment was repeated five times with similar results. Data for ERF55 are shown in Supplementary Fig. 13c. **g** ChIP-qPCR. Germinating seeds of Col-0 *p35S:HA-YFP-ERF58* (line 21C) and Col-0 *p35S:YFP-HA* (negative control) were subject to light treatments as described in Fig. 4a. Chromatin was isolated and HA-YFP-ERF58 bound DNA fragments were purified using α-GFP coupled magnetic beads. qPCR and specific primers were used to detect the *GA2ox4* −1111...−1264 promoter fragment in the eluate fraction. *ACT7* was used for normalisation. Values show enrichment of the *GA2ox4* promoter fragment in the eluate fraction compared to the input fraction and are means of three replicates ±SD. **a–c, g** Different letters indicate significant differences as determined by two-way ANOVA followed by post-hoc Tukey's HSD test. **a–c** $p < 0.05$. **g** $p < 0.01$.

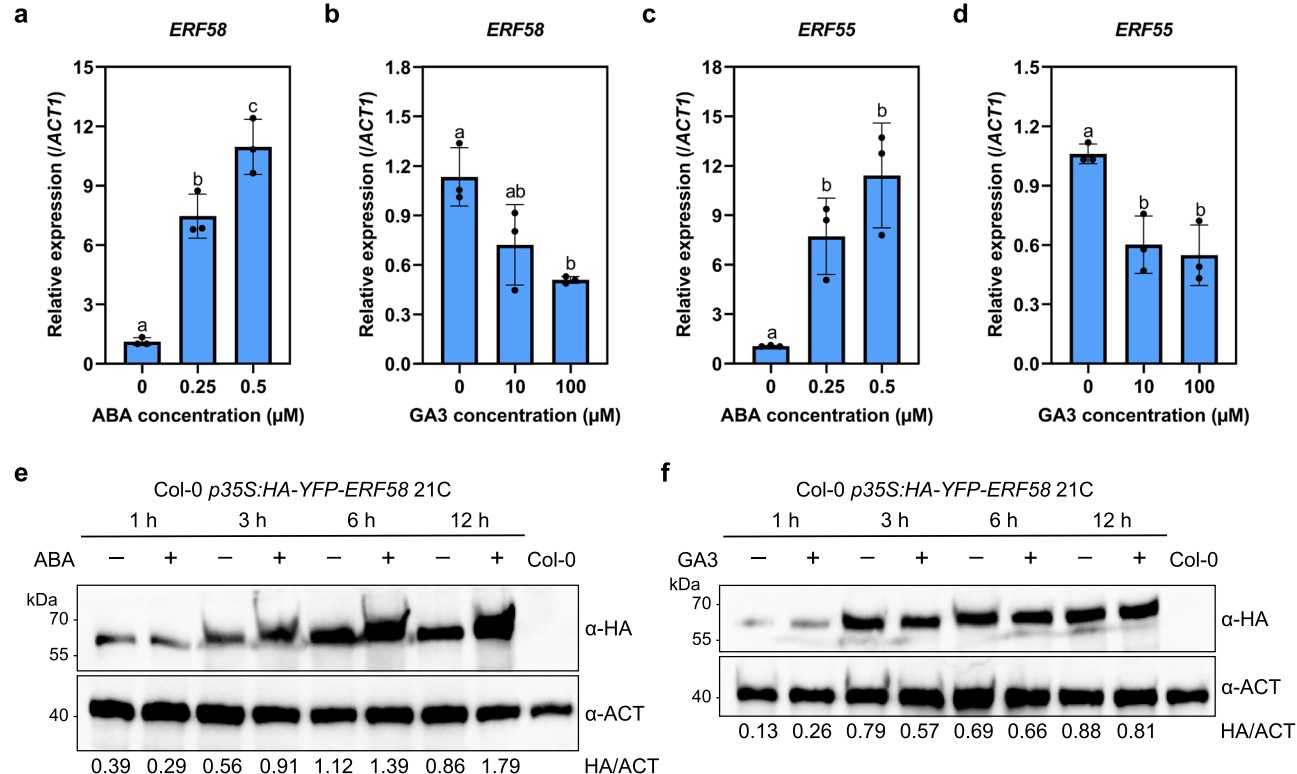

**Fig. 8 ABA and GA regulate *ERF55* and *ERF58* expression. a–d** RT-qPCR analysis of *ERF58* (**a**, **b**) and *ERF55* expression (**c**, **d**) in seeds incubated in the dark for 12 h on different concentrations of ABA (**a**, **c**) or GA3 (**b**, **d**). *ACT1* was used as an internal control. Values show means of three replicates ±SD. Different letters indicate significant differences as determined by one-way ANOVA followed by post-hoc Tukey's HSD test; $p < 0.05$. **e**, **f** Effect of ABA and GA3 on HA-YFP-ERF58 protein levels in ERF58ox seeds. Col-0 *p35S:HA-YFP-ERF58* (line 21C) seeds were incubated in the dark on 0.5 µM ABA (**e**) or 100 µM GA3 (**f**) for 1, 3, 6, or 12 h. Control seeds were incubated on water. Total protein was extracted and analysed by SDS-PAGE and immunoblotting with α-HA; α-ACT was used to detect ACTIN as loading control. Col-0 was included as negative control. Signals detected by α-HA and α-ACT were quantified using ImageJ; numbers below the membrane show the HA/ACT ratio. The experiments were repeated three times with similar results.

groups of the AP2/ERF family and put them under light control, offering huge potential for signal integration between light signalling and different signalling pathways controlled by AP2/ERFs.

Regulation of completion of seed germination is highly complex with different environmental cues that have to be integrated[2,3]. Such multiple input/single output regulation is hardly possible with a simple linear pathway and requires complex regulatory networks. We identified ERF55 and ERF58 as additional nodes in the regulatory network controlling the completion of seed germination (Fig. 9b). The effect of removing a component from such a network is difficult to predict, since it is usually not known to what extent other parts of the network can compensate and to what extent the signal can be rerouted and take other paths through the network. Although we cannot yet fully answer these questions for the *erf55 erf58* mutant, we have identified several genes encoding ABA anabolic or GA catabolic enzymes that are likely direct targets of ERF55 and ERF58, i.e. they are upregulated by ERF55 and ERF58, they contain a DRE element as potential ERF55/ERF58 binding site in their promoter, and both ERF55 and ERF58 associate with their promoter in ChIP-qPCR experiments and/or EMSA. In contrast, none of these genes has been identified as direct target of PIF1 or SOM; in a ChIP-chip approach, not a single gene encoding ABA anabolic or GA catabolic enzymes has been identified as direct target of PIF1, and the mechanism by which SOM regulates gene expression is still unclear[7,12,13]. Here, however, we show that ERF55 and ERF58 have the potential to directly link light activation of phytochromes to regulation of a set of genes that encode ABA anabolic or GA catabolic enzymes.

Feedback regulation adds another layer of complexity to the regulatory network controlling the completion of seed germination. We found that ERF55 and ERF58 not only regulate genes encoding ABA or GA metabolic enzymes, but are themselves regulated by GA and in particular ABA, with GA reducing expression of *ERF55* and *ERF58*, and ABA promoting it. There is evidence that *PIF1* and *SOM* also are subject to positive feedback regulation by GA and ABA[33–35], which together with feedback regulation of *ERF55* and *ERF58* could insure rapid promotion of germination completion under favourable conditions and reliable inhibition of the completion of germination when conditions are unfavourable.

Taken together, we identified ERF55 and ERF58 as nodes in the regulatory network controlling the completion of seed germination. We propose a model in which light-activated phyA and phyB directly interact with ERF55 and ERF58 to prevent their association with the promoter of genes repressing germination completion. PhyA and phyB also reduce the expression of *ERF55* and *ERF58* in seeds exposed to light, thereby further promoting germination completion under phyA/phyB-ON conditions. The *erf55-1 erf58-2* double mutant has a percentage of seeds capable of completing germination in the absence of light, highlighting the function of ERF55 and ERF58 in light regulation of the completion of seed germination and supporting the model proposed in Fig. 9.

Due to extensive redundancy, AP2/ERFs may not have been identified as light signalling components in classical loss-of-function screens. However, the fact that phyA and phyB bind to the AP2 domain of ERF58, which is highly conserved in all

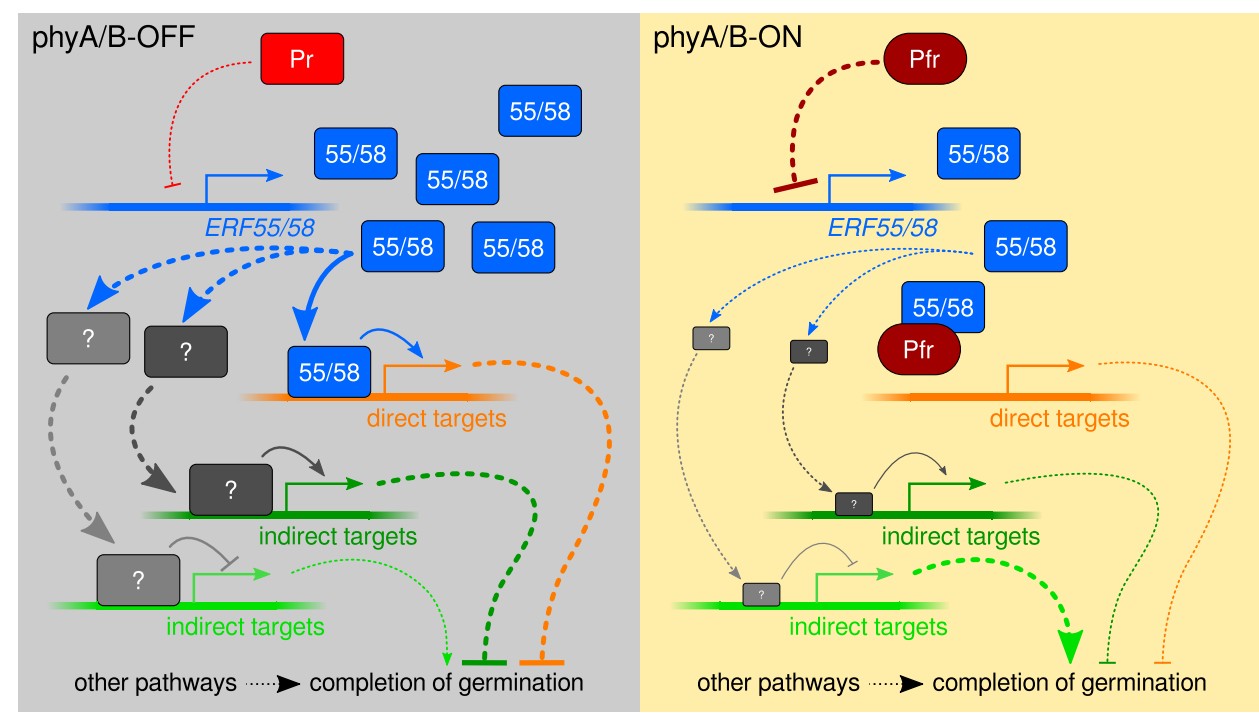

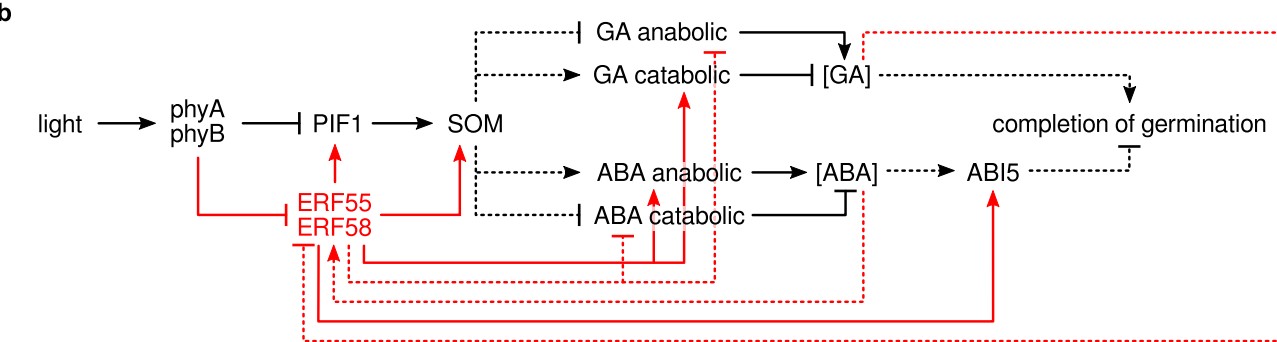

**Fig. 9 Light regulation of the completion of seed germination by ERF55 and ERF58. a** Model for regulation of the completion of seed germination by ERF55/ERF58 and phytochromes. In phyA/B-OFF conditions (e.g. in the dark), ERF55 and ERF58 bind to the promoter of genes that repress the completion of seed germination and enhance their expression (direct targets; shown in orange). Genes that promote the completion of seed germination can be repressed by ERF55/ERF58 through upregulation of intermediate factors (shown in grey) but they are possibly not direct targets of ERF55/ERF58. Indirect targets of ERF55/ERF58 are shown in green and may either promote (light green) or repress (dark green) completion of germination. Light-activated phytochromes repress the expression of *ERF55* and *ERF58* in phyA/B-ON conditions and also bind ERF55 and ERF58 to prevent their association with the promoter of various target genes that repress the completion of germination. Solid lines indicate direct regulation; dashed lines indicate indirect regulation; →, positive regulation; ⊣, negative regulation; thick arrows, strong regulation; thin arrows, week or no regulation. **b** Simplified regulatory network for light regulation of the completion of seed germination. Regulatory interactions of ERF55/ERF58 are shown in red; other regulatory interactions are shown in black. Solid lines indicate direct regulation; dashed lines indicate indirect regulation or regulatory steps for which it is not known whether they are direct or indirect; →, positive regulation; ⊣, negative regulation.

AP2/ERF transcription factors, opens the possibility that responses mediated by a wide range of AP2/ERFs could be under light control by phytochromes.

## Methods

**Vectors and cloning**. Details of plasmid constructs and primers used in this study can be found in Supplementary Tables 1, 2, and Supplementary Fig. 16.

**Plant material and growth conditions**. All Arabidopsis lines used in this study are in the *Arabidopsis thaliana* Columbia (Col-0; NASC N1092) background. The *phyB-9* (NASC N6217)[36], *phyA-211* (NASC N6223)[37], *som-3* (SALK_008075, NASC N508075)[13], and Col-0 *p35S:SOM-FLAG*[34] have been described previously. *erf55-1* (SALK_091212, NASC N681465), *erf56* (SALK_139786, NASC N661174), *erf57* (GK-098F06, NASC N409378), *erf58-1* (SALK_139727, NASC N677156), *erf58-2* (SAIL_1293_A03, NASC N848391), *erf59* (SALK_020767, NASC N654090), *erf60* (SALK_021999, NASC N656258), *pif1-1* (SAIL_256_G07,

NASC N66041), and *abi5-8* (SALK_013163C, NASC N673861) were obtained from the NASC[38–40]. Col-0 *pERF58:HA-YFP-ERF58*, Col-0 *p35S:HA-YFP-ERF58* (ERF58ox), Col-0 *p35S:ABI5-YFP-HA*, Col-0 *p35S:HA-YFP-PIF1*, *erf55-1 pER-F55:ERF55*, *erf55-1 erf58-2 p35S:ABI5-YFP-HA*, and *erf55-1 erf58-2 p35S:HA-YFP-PIF1* were created by using the floral dip method[41]. The Col-0 *p35S:YFP-HA* line was described previously[25]. The *erf55-1 erf58-2 phyA-211*, *erf55-1 erf58-2 phyB-9*, *erf55-1 erf58-2 pif1-1*, *erf55-1 erf58-2 abi5-8*, *erf55-1 erf58-2 p35S:SOM-FLAG*, and *erf58-2 pERF58:HA-YFP-ERF58* lines were generated by genetic crossing.

For propagation, Arabidopsis plants were grown in the greenhouse. Tobacco (*Nicotiana benthamiana*) plants were cultivated under 14 h day (26 °C)/10 h night (19 °C) cycles.

Genotyping was performed as described in Supplementary Table 3 using primers listed in Supplementary Table 4.

**Germination assay**. Seeds used in the same experiment were obtained from plants grown at the same time in the same conditions and harvested at the same time. Differences during plant cultivation and seed set resulted in seed batches with

higher or lower dormancy. Final cumulative germination percentage under different treatments or of different genotypes is therefore comparable within the same experiment but not between different experiments. Seeds were after-ripened for at least one month at room temperature before using them for germination assays.

Final cumulative germination percentage was quantified as follows[11,42]. Sixty seeds were surface sterilised and sown on filter paper (Macherey-Nagel; Cat. no. MN 615) soaked with 4.5 ml ddH$_2$O or on ½× MS plates/1.5% agar. For testing phyB-dependent completion of seed germination, plates were placed in the dark at 22 °C for one hour and then exposed to either 3.82 µmol m$^{-2}$ s$^{-1}$ FR light for 5 min (phyB-OFF), or 3.82 µmol m$^{-2}$ s$^{-1}$ FR light for 5 min followed by 2.5 µmol m$^{-2}$ s$^{-1}$ R light for 5 min (phyB-ON). Plates were then incubated for 5 days in the dark at 22 °C before evaluating seed germination percentages. For testing phyA-dependent completion of germination, seeds were sown on filter paper (Macherey-Nagel; Cat. no. MN 615) soaked with 4.5 ml ddH$_2$O or on ½× MS plates/1.5% agar, incubated in the dark at 22 °C for one hour, exposed to 3.82 µmol m$^{-2}$ s$^{-1}$ FR light for 5 min, and then either kept in the dark for 7.5 days (phyA-OFF) or kept in the dark for two days, exposed to 3.82 µmol m$^{-2}$ s$^{-1}$ FR light for 12 h, and kept in the dark for another 5 days (phyA-ON). Seed germination percentage was then quantified. To confirm viability of seeds, they were sown on filter paper soaked with ddH$_2$O or on ½× MS plates/1.5% agar and exposed to W light for 5 or 7.5 days.

To evaluate completion of seed germination in presence of ABA or PAC, 60 surface-sterilised seeds were sown on filter paper (Macherey-Nagel; Cat. no. MN 615) soaked with 4.5 ml ddH$_2$O supplemented with different concentrations of ABA (Cayman Chemical Company, Michigan, USA; Cat. no. 10073) or PAC (Sigma-Aldrich, Cat. no. 46046), exposed to W light for 6 h, and then incubated in the dark at 22 °C for 7 days before scoring for completion of germination. To evaluate completion of seed germination in presence of GA, 60 surface-sterilised seeds were sown on filter paper (Macherey-Nagel; Cat. no. MN 615) soaked with 4.5 ml ddH$_2$O supplemented with different concentrations of GA3 (Duchefa Biochemie B.V, Haarlem, Netherlands, Cat. no. G0907) and incubated in the dark at 22 °C for 7 days before scoring for completion of germination.

All seed germination experiments have at least three biological replicates.

**Transient transformation and promoter activity assays.** Agrobacterium C58 carrying the respective plasmids were co-infiltrated into leaves of 3–5-week-old *N. benthamiana* plants according to Grefen et al.[43]. To suppress transgene silencing, Agrobacteria containing a plasmid coding for the p19 protein from tomato bushy stunt virus was co-infiltrated[44]. After infiltration, the plants were incubated in W light for 16 h and then transferred to the dark at 22 °C for 2–3 days. The localisation of the fusion proteins in transformed epidermal leaf cells of the *N. benthamiana* plants was then analysed using an Axioplan 2 microscope (Zeiss, Oberkochem, Germany) equipped with a Photometrics CoolSNAP HQ CCD Monochrome camera and filter sets for YFP (F31-028, excitation 500 nm, emission 515 nm; AHF Analysentechnik, Tübingen, Germany), and CFP (F31-044, excitation 436 nm, emission 455 nm; AHF Analysentechnik, Tübingen, Germany). For promoter activity assays, we used the promoters to be tested fused to the CDS of luciferase as reporter, *p35S:HA-YFP-ERF58*, *p35S:HA-YFP-ERF55*, *p35S:ABI5-YFP-HA*, *p35S:HA-YFP-PIF1*, and *p35S:HA-YFP* (empty vector, negative control) as effectors, and *p35S:GUS* as an internal control for normalisation. Total protein was then extracted from leaf discs of infiltrated leaves using LUCI buffer (100 mM K$_2$PO$_4$ pH 7.8, 0.05% [v/v] Tween 20, protease inhibitor cocktail [Sigma-Aldrich; Cat. no. P2714], 20 mM MG132 [Sigma-Aldrich; Cat. no. 474790], and 1 mM DTT)[45]. Luciferase luminescence and GUS fluorescence was then measured according to Xu et al.[45]. The relative luciferase activity was calculated as ratio of luciferase luminescence/GUS fluorescence. Constructs used for the assays and primers for cloning the constructs are described in Supplementary Tables 1 and 2.

**Yeast interaction assay.** Yeast two-hybrid assays (Y2H) were performed as described[22] using the plasmids listed in Supplementary Table 1. Briefly, we co-transformed plasmids into *Saccharomyces cerevisiae* strain NMY51[46] for growth assays and for liquid quantitative ortho-nitrophenyl-β-galactoside (ONPG) assays using the Frozen-EZ yeast transformation kit (Zymo Research, Freiburg, Germany; Cat. no. T2001). For selection, transformed yeast were grown at 28 °C on CSM/2% agar plates lacking leucine and tryptophan (CSM -L-T). For Y2H growth assays, 3–10 colonies were resuspended in sterile water. OD$_{600}$ was adjusted to 0.1 and 5 µl of resuspended cultures were then spotted onto CSM/2% agar plates lacking leucine, tryptophan, and histidine (CSM -L-T-H), and grown at 28 °C until colonies were visible. For assays with phyB, CSM -L-T-H plates were supplemented with 5 mM 3-amino-1,2,4-triazole (Sigma-Aldrich; Cat. no. A8056). For Y2H assays with phyA or phyB, CSM -L-T-H plates were supplemented with 20 µM phycocyanobilin (PCB) (Livchem Logistics, Frankfurt, Germany; Cat. no. FSIP14137) and incubated at 26 °C in either darkness or R light (2 µmol m$^{-2}$ s$^{-1}$). For ONPG assays, yeast was grown over night in CSM medium lacking leucine and tryptophan (CSM -L-T) supplemented with 20 µM PCB (Livchem Logistics, Frankfurt, Germany; Cat. no. FSIP14137). Yeast cultures were then supplemented with 1.5 ml YPDA and exposed to 10 µmol m$^{-2}$ s$^{-1}$ FR light for 5 min either followed by 10 µmol m$^{-2}$ s$^{-1}$ R light for 5 min or not and incubated for another 3–4 h at 26 °C in the dark. Yeast was then harvested in safe green light and β-Gal activity was quantified using an ONPG assay according to the Clontech Y2H manual[47].

**Quantification of transcript levels by RT-qPCR.** Total RNA was extracted using a modified version of the method described by Malnoy et al.[48]. One hundred mg seeds were ground to fine powder in liquid nitrogen using a mortar and pestle, and extracted in 1 ml of extraction buffer (140 mM LiCl, 100 mM Tris-HCl pH 8, 10 mM EDTA, 5% SDS). The samples were gently mixed and centrifuged at 15,100 × g for 15 min at 4 °C. The supernatant was then transferred to a new tube and extracted with an equal volume of chloroform:isoamyl alcohol (24:1). After centrifugation (15 min, 15,100 × g, 4 °C), the supernatant was transferred to a new tube containing 1/3 volume 5 M KAc (pH 5.5), gently mixed, and extracted again with chloroform:isoamyl alcohol (24:1). The supernatant was then transferred to a new tube containing 0.625 volumes of 8 M LiCl, gently mixed, and incubated at −20 °C for 6–8 h, followed by further clean-up using the ISOLATE II RNA Plant kit (Bioline Meridian Bioscience, London, UK; Cat. no. BIO-52077). Reverse transcription into cDNA was performed using the High Capacity Reverse Transcription kit (Thermo Fisher Scientific, Waltham, MA, USA; Cat. no. 4368814), and qPCR was carried out using the SensiFAST™ SYBR Hi-ROX kit (Bioline Meridian Bioscience, London, UK; Cat. no. BIO-92005). Each experiment was performed in triplicate with three technical replicates. Gene expression levels were normalised to *ACT1* (At1g32200) or *ACT7* (At5g09810). The primers used for RT-qPCR can be found in Supplementary Table 5.

**Protein expression and purification.** For expression of MBP, MBP-ERF58, MBP-ERF55, and MBP-FHY1 (163–202), the plasmids *pMAL*, *pMAL:ERF58*, *pMAL:ERF55*, or *pMAL:FHY1 (163–202)* were transformed into the *E. coli* strain BL21(DE3)-RIL. For expression of MBP, MBP-ERF55, and MBP-ERF58, transformed cells were inoculated into 10 ml LB medium supplemented with 100 µg ml$^{-1}$ Ampicillin (Duchefa; Cat. no. A0104), grown overnight at 37 °C, and diluted into 1 l LB medium/100 µg ml$^{-1}$ Ampicillin. This culture was then grown at 37 °C until OD$_{600}$ = 0.6 and induced with 0.4 mM IPTG (Isopropyl-1-thio-β-D-galactopyranoside) (Carl-Roth, Karlsruhe, Germany; Cat. no. 2316.3). The induced culture was grown overnight at 18 °C, and centrifuged at 3800 × g for 30 min at 4 °C. The cell pellet from 1 l culture was resuspended in 20 ml PBS supplemented with 0.5 mM EDTA and 1 mM PMSF, and Lysozyme (Carl-Roth, Karlsruhe, Germany; Cat. no. 8259.1) was added to a final concentration of 1 mg ml$^{-1}$. After stirring for 30 min on ice, the cells were lysed by ultra-sonication on ice (three times for 2 min). The cell lysate was centrifuged for 30 min at 4 °C at maximum speed. The supernatant was then applied to the amylose resin and the resin was washed according to the manufacturer's instruction (NEB, Ipswich, MA, USA; Cat. no. E8021L). Proteins were eluted with elution buffer (20 mM Tris-HCl pH 7.4, 0.2 M NaCl, 1 mM EDTA, 10 mM maltose).

Expression of MBP-FHY1 (163–202) for subsequent affinity purification of phyA was done as described above with the following modifications. IPTG was added to 2 mM final concentration, induction was performed at 37 °C for 2 h, and pelleted bacteria were resuspended in 45 ml 50 mM KPO$_4$ pH 7.8, 20 mM NaCl, 1 mM EDTA, 2 mM DTT, protease inhibitors (Sigma-Aldrich; Cat. no. I3911; 1 vial per litre), and subsequently divided into 5× 10 ml and frozen at −20 °C until the day of phyA purification.

For expression of photoactive phyA-FLAG in *E. coli*, *pDS15A-PHYA-FLAG-PϕB MKII* was transformed into the *E. coli* strain BL21(DE3)-RIL. Transformed cells were inoculated into 10 ml LB medium supplemented with 30 µg ml$^{-1}$ Kanamycin (Duchefa; Cat. no. K0126), grown at 28 °C for 24 h, and diluted into 1 l Terrific Broth (25 g l$^{-1}$ yeast extract, 10 g l$^{-1}$ tryptone, 10 g l$^{-1}$ peptone, 1.2% [v/v] glycerol, 100 mM KPO$_4$ pH 7.4)/30 µg ml$^{-1}$ Kanamycin. This culture was grown at 28 °C for 5 h until OD$_{600}$ > 1. The temperature was then decreased to 16 °C and 50 ml 100 mM Malate/100 mM Glutamate solution was added to the cell culture. After 1 h, the culture was induced with 2 mM IPTG, covered with black cloth and incubated overnight at 16 °C in the dark. The cell culture was then centrifuged at 3800 × g for 30 min at 4 °C. The cell pellet was resuspended in 45 ml resuspension buffer (50 mM KPO$_4$ pH 7.8, 20 mM NaCl, 1 mM EDTA, 2 mM DTT, 0.1% [v/v] NP-40, 2.5% [v/v] ethylene glycol, protease inhibitors [Sigma-Aldrich; Cat. no. I3911, 1 vial per litre]). Lysis was performed by adding Lysozyme (Sigma-Aldrich; Cat. no. L6876) to 0.3 mg ml$^{-1}$ and incubating at room temperature for 15 min until viscous. DNA was then degraded by first adding MgCl$_2$ to 5 mM and CaCl$_2$ to 1 mM, then adding DNase I (Sigma-Aldrich; Cat. no. DN25) to 0.02 mg ml$^{-1}$ and incubating for 15 min at room temperature. The lysate was then cleared by centrifugation at 20,000 × g, at 4 °C for 30 min, then EDTA was added to 10 mM to the supernatant to inactivate the DNase I, followed by centrifugation at 20,000 × g, at 4 °C for 30 min; the supernatant was then passed through a 0.45 µm filter. All following steps were performed in complete darkness at 4 °C with minimal use of safe green light. The cell lysate containing phyA-FLAG was irradiated with strong R light (655 nm LED, 200 µmol m$^{-2}$ s$^{-1}$) for 5 min to activate phyA and then covered with foil. The lysate was then loaded at 1 ml min$^{-1}$ onto a 5 ml amylose resin (NEB, Ipswich, MA, USA; Cat. no. E8021L) column that had been preloaded with MBP-FHY1 (163–202), washed (50 mM KPO$_4$ pH 7.8, 20 mM NaCl, 1 mM EDTA) and covered with foil. As phyA in the Pfr form interacts with the C-terminus of FHY1, it is captured on the column. The column was then washed with 50 mM KPO$_4$ pH 7.8, 20 mM NaCl, 1 mM EDTA at 1 ml min$^{-1}$ for 1 h (60 ml). The flow was then stopped, and the column irradiated with strong FR light (740 nm LED, 200 µmol m$^{-2}$ s$^{-1}$) for 5 min to convert phyA to the Pr form, releasing it from FHY1. Continuing to irradiate with FR light, phyA-FLAG was

eluted at 1 ml min⁻¹. Fractions containing protein as determined by nanodrop (Thermo Fischer Scientific) were pooled, DTT added to 2 mM and Protease inhibitors (Sigma-Aldrich; Cat. no. I3911) added, followed by concentration using a vivaspin 6 ml centrifugal concentrator (100 kDa cut-off PES, prewashed with ddH$_2$O and column wash buffer; Sartorius; Cat. no. VS0642) at $4000 \times g$, 4 °C for 10 min.

**Electrophoretic mobility shift assay**. Electrophoretic mobility shift assays (EMSAs) were done according to the LightShift Chemiluminescence EMSA kit manual (Thermo Fisher Scientific, Waltham, MA, USA; Cat. no. 20148) with modifications. Biotinylated probes were used in this study. Biotin-labelled wild-type probes were obtained by PCR or by annealing complementary oligonucleotides (Supplementary Table 6) using primers listed in Supplementary Table 7. Biotin-labelled DRE mutant probes were obtained by overlap extension PCR with the mutated DRE element included in the overlap (Supplementary Table 6). Cold probes consisted of annealed complementary oligonucleotides DS979 and DS980 (Supplementary Table 7); they were annealed by heating to 95 °C for 5 min followed by 5 min incubation of ice. First, unlabelled DNA (cold competitor, 5 μM final concentration) and proteins (5 μg MBP-ERF55 or MBP-ERF58; 10 μg phyA-FLAG if applicable), if applicable, were pre-incubated in 20 μl 1× binding buffer (10 mM Tris-HCl pH 8.0, 150 mM KCl, 0.5 mM EDTA pH 8.0, 0.1% Triton-X 100, 12.5% glycerol, 0.2 mM DTT [freshly added before use]) supplemented with 0.25 mg ml⁻¹ BSA (NEB, Cat. no. B9001S) at 24 °C for 20 min. Then biotin-labelled probes (50 nM final concentration) were added to the respective binding reactions followed by incubation at 24 °C for 20 min. Samples were analysed on 6% non-denaturing polyacrylamide gels, which were run in 0.5× Tris-Borate-EDTA buffer at 100 V. Gels were then blotted onto a nylon membrane at 380 mA (~100 V) for 30 min and DNA was immediately crosslinked to the membrane at 120 mJ cm⁻² using a UV crosslinker. The membrane was then blocked overnight at 4 °C in 30 ml 0.5% BSA buffer and incubated under gentle shaking for 15 min at room temperature in 0.5% BSA buffer containing 1:500,000 diluted Stabilised Streptavidin-Horseradish Peroxidase Conjugate (Thermo Fisher Scientific, Waltham, MA, USA; Cat. no. 21126). After washing the membrane twice or three times with PBST and once with PBS, Amersham ECL Prime Western Blotting Detection Reagent (GE Healthcare, Chicago, IL, USA; cat. no RPN2232) was spread on the membrane and luminescence signals were detected using a CCD camera.

**Zinc blot assay**. Zinc blot assays were performed as described in Kami et al.[49] with slight modifications. Samples prepared for the EMSA experiment were run on native gels containing 1 mM Zinc-acetate in running buffer supplemented with 1 mM Zinc-acetate. Fluorescence signals (caused by complex formation of zinc ions and the chromophore of phytochromes) were detected on a UV-illuminator using a camera equipped with a red-light filter.

**In vivo co-immunoprecipitation assay**. Six hundred μl Col-0 *pERF58:HA-YFP-ERF58* seeds were sown on different ½× MS plates/1.5% agar. The plates were incubated for 2 days at 4 °C in the dark, exposed for 8 h to 10 μmol m⁻² s⁻¹ R light (660 nm) at 22 °C to induce germination, returned to the dark, and incubated for another 64 h at 22 °C. Then, seedlings on different plates were exposed to different light treatments (at 22 °C): D, plates were kept in the dark; 5' R, plates were exposed to 20 μmol m⁻² s⁻¹ R light for 5 min; 2 h FR, plates were exposed to 10 μmol m⁻² s⁻¹ FR light for 2 h; 2 h FR + 5' R, plates were exposed to 10 μmol m⁻² s⁻¹ FR light for 2 h followed by 20 μmol m⁻² s⁻¹ R light for 5 min. Total protein was then extracted and used for co-immunoprecipitation. Native extraction buffer contained 100 mM phosphate buffer 7.8, 150 mM NaCl, 1 mM KCl, 1 mM EDTA pH 8.0, 1% PEG 4000, 0.5% Triton X-100, 50 mM MG132 (Sigma-Aldrich; Cat. no. 474790), 1× Protease inhibitor Cocktail (Sigma-Aldrich; Cat. no. I3911), and 1× cOmplete Protease Inhibitor Cocktail (Sigma-Aldrich; Cat. no. 04693159001). For each sample, 50 μl of α-GFP MicroBeads (Miltenyi Biotec, Bergisch Gladbach, Germany; Cat. no. 130-091-125) were added to the extracts, followed by 2 h of incubation in darkness with gently shaking. Immunoprecipitated complexes were washed twice with native extraction buffer and three times with native extraction buffer without KCl, and eluted in 2× SDS loading buffer (100 mM Tris-HCl pH 6.8, 4% SDS, 20% glycerol, 0.05% bromophenol blue). Samples in 2× SDS loading buffer were incubated at 80 °C for 5 min and analysed by SDS-PAGE and immunoblotting. α-phyA (Agrisera, Vännäs, Sweden; Cat. no. AS07 220; polyclonal, rabbit, dilution 1:1,500), α-phyB (monoclonal, mouse, B6-B3, dilution 1:250)[50], and α-HA (BioLegend, San Diego, CA, USA; Cat. no. 901502; monoclonal, mouse, 16B12, dilution 1:2,000) antibodies were used to detect proteins. Alkaline phosphatase goat α-rabbit IgG antibody (Vector Laboratories, Burlingame, CA, USA; Cat. no. AP-1000; dilution 1:7,500) or alkaline phosphatase horse α-mouse IgG antibody (Vector Laboratories, Burlingame, CA, USA; Cat. no. AP-2000; dilution 1:10,000) was used as secondary antibody with CDP-Star (Sigma-Aldrich; Cat. no. 11759051001) as substrate for signal detection.

**Seed protein extraction and immunoblotting**. Extraction and acetone precipitation of seed proteins for immunoblot analysis was performed according to Piskurewicz and Lopez-Molina[51] with slight modifications. Briefly, 100 mg seeds were ground to a fine powder in liquid nitrogen using a mortar and pestle,

extracted in 400 μl of extraction buffer (4% SDS, 2% β-mercaptoethanol, 20% glycerol, 100 mM Tris-HCl pH 8.0), vortexed thoroughly, and boiled at 95 °C for 3 min. Extracts were then centrifuged at $15{,}100 \times g$ for 10 min at room temperature. The supernatant was collected and transferred to a clean 2 ml microcentrifuge tube containing four volumes of acetone. After careful mixing, proteins were precipitated for 20 min at −20 °C. Samples were then centrifuged at $15{,}100 \times g$ for 10 min at 4 °C. The supernatant was discarded and the pellet was washed with 80% acetone and resuspended in 150 μl SDS loading buffer (100 mM Tris-HCl pH 6.8, 4% SDS, 20% glycerol, 0.05% bromophenol blue). Samples were then analysed by SDS-PAGE and immunoblotting. α-HA and α-actin (Sigma-Aldrich, St. Louis, MO, USA; Cat. no. A0480, monoclonal, mouse; 1:3,000 dilution) antibodies were used to detect proteins. Blots were quantified according to Enderle et al.[23] using ImageJ.

**ChIP assay**. ChIP assays were performed based on the protocol by Yamaguchi et al.[52] with some modifications. Col-0 *p35S:HA-YFP-ERF58* and Col-0 *p35S:YFP-HA* stable transgenic lines were used for ChIP. One gram (corresponding to ~1.5 ml) of seeds germinating under conditions described in the respective figures were cross-linked with 3% formaldehyde solution in PBS under a vacuum for 1 h. Then, 2.5 ml 2 M Glycine were added and vacuum was applied for another 5 min. Chromatin was then extracted (extraction buffer: 100 mM MOPS pH 7.6, 10 mM MgCl$_2$, 5% Dextran T-40, 2.5% Ficoll 400, 0.5% [w/v] BSA [IgG free], 10 mM DTT, 1× cOmplete Protease Inhibitor Cocktail [Sigma-Aldrich; Cat. no. 04693159001], 50 μM MG132 [Sigma-Aldrich; Cat. no. 474790], 0.4 M Sucrose), sheared to an average length of 500–1000 bp by sonication, and then immunoprecipitated with 100 μl α-GFP MicroBeads (Miltenyi Biotech; Cat. no. 130-091-125) for 2 h in darkness with gently shaking. Immunoprecipitated complexes were eluted in 200 μl nuclei lysis buffer (50 mM Tris-HCl pH 8.0, 10 mM EDTA pH 8.0, 1% SDS) heated to 65 °C. Immunoprecipitated DNA fragments were then obtained by DNA reverse cross-linking and used for qPCR with gene-specific primers (Supplementary Table 8). *ACT7* was used for normalisation; enrichment of promoter fragments in the eluate fraction was calculated as abundance in the eluate fraction divided by abundance in the input fraction.

**Measurement of ABA content in germinating seeds**. Dry seeds (Col-0, *erf55-1 erf58-2*, *ERF58ox* 21C; 3 replicates per genotype; 1.5 g dry seeds per replicate) were incubated on ddH$_2$O for 24 h. Excess water was drained off, seeds were frozen in liquid nitrogen, and stored at −80 °C until shipment to collaborators performing the ABA measurement. The ABA content in the seeds was measured as follows. The tubes containing the frozen germinating seeds were defrosted at 10 °C in an ultrasonic bath (2× 10 min). For each replicate, 2× 100 mg (±10%) germinating seeds were weighed and transferred into separate 2 ml safelock tubes containing a 5 mm steel ball. The seeds were frozen in liquid nitrogen and ground to powder using a Retch mill (4× 15 s; 25 Hz) with intermittent cooling in liquid nitrogen. All solvents used in the following steps were prechilled at 8 °C. Three hundred sixty μl MeOH (containing 100 nM D6-ABA) were added to the ground material. The samples were mixed and incubated for 1 min in an ultrasonic bath at 10 °C and then for 10 min at 10 °C in a thermoshaker at 950 rpm. After adding 200 μl CHCl$_3$, the samples were incubated again for 1 min in an ultrasonic bath at 10 °C followed by 10 min at 10 °C in a thermoshaker at 950 rpm. Then, 400 μl water were added to the samples before incubating them again in an ultrasonic bath and thermomixer under the above-described conditions. After 10 min centrifugation at 10 °C, $18{,}600 \times g$, 20 μl supernatant were diluted 1/5 with water and measured using targeted LCMS.

The LCMS profiling analysis was performed using a Micro-LC M5 (Trap and Elute) and a QTRAP6500 + (Sciex) operated in MRM mode (MRMs ABA (1) quantifier ion (m/z) Q1/Q3 263/153, declustering potential DP −40 V, collision energy CE −15V; ABA (2) (m/z) Q1/Q3 263/219, DP −40 V, CE −20V; D6-ABA (1) quantifier ion (m/z) Q1/Q3 269/159; D6-ABA (2) (m/z) Q1/Q3 269/225). Chromatographic separation was achieved on a Luna Omega Polar C18 column (3 μm; 100 Å; 150× 0.3 mm; Phenomenex) and a Luna C18(2) trap column (5 μm; 100 Å; 20× 0.3 mm; Phenomenex) with a column temperature of 35 °C. The following binary gradient was applied for the main column at a flow rate of 10 μl min⁻¹: 0–0.2 min, isocratic 90% A; 0.2–2 min, linear from 90% A to 30% A; 2–4.5 min, linear from 30% A to 10% A; 4.5–5 min, linear from 10% A to 5% A; 5–5.3 min, isocratic 5 % A; 5.3–5.5 min, linear from 5% A to 90% A; 5.5–6 min, isocratic 90% A (A: water, 0.1% aq. formic acid; B: acetonitrile, 0.1% aq. formic acid). The samples were then concentrated on the trap column using the following conditions: flow rate 50 μl min⁻¹: 0–1.5 min isocratic 95% A; 1.5 min start main gradient; 1.5–1.7 min isocratic 95% A. The injection volume was 50 μl. Analytes were ionised using an Optiflow Turbo V ion source equipped with a SteadySpray T micro electrode (10–50 μl min⁻¹) in positive (ion spray voltage: 4800 V) and negative (ion spray voltage: −4500 V) ion mode. Following additional instrument settings were applied: nebuliser and heater gas, nitrogen, 25 and 45 psi; curtain gas, nitrogen, 30 psi; collision gas, nitrogen, medium; source temperature, 200 °C; entrance potential, ±10 V; collision cell exit potential, ±25 V; scan time 5 ms. The ABA content in a replicate was calculated as the mean of the two samples taken from the respective replicate. Three biological replicates were measured.

**Statistics**. Details are described in the respective figure legends and the source data file. Exact p-values are included in the source data file.

**Accession numbers**. *AAO3*, At2g27150; *ABA1*, At5g67030; *ABA2*, At1g52340; *ABI5*, At2g36270; *ACT1*, At1g32200; *ACT7*, At5g09810; *CYP707A2*, At2g29090; *ERF53*, At2g20880; *ERF54*, At4g28140; *ERF55*, At1g36060; *ERF56*, At2g22200; *ERF57*, At5g65130; *ERF58*, At1g22190; *ERF59*, At1g78080; *ERF60*, At4g39780; *ERF61*, At1g64380; *ERF62*, At1g13620; *GA2ox2*, At1g30040; *GA2ox4*, At1g47990; *GA3ox1*, At1g15550; *GA3ox2*, At1g80340; *NCED6*, At3g24220; *NCED9*, At1g78390; *PHYA*, At1g09570; *PHYB*, At2g18790; *PIF1*, At2g20180; *SOM*, At1g03790.

**Reporting summary**. Further information on research design is available in the Nature Research Reporting Summary linked to this article.

## Data availability

All data generated or analysed during this study are included in this published article (and its supplementary information files). Source data are provided with this paper.

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

## Acknowledgements

This study was supported by the German Research Foundation (DFG) under Germany's Excellence Strategy (EXC-2189 – Project ID 390939984, project C1 to AH) and by a DFG research grant to AH (HI 1369/9-1, ID 445757564). ZL was supported by a CSC fellowship (CSC no. 201606910040). Plant hormone analyses were additionally funded by the Deutsche Forschungsgemeinschaft (DFG, German Research foundation) – Projektnummer 442641014. We are grateful to M. Krenz and T. Albonetti (University of Freiburg, Germany) for technical assistance, Dr. William Teale and Dr. Dorothee Lambert (University of Freiburg, Germany) for critically reading the manuscript, Prof. Giltsu Choi (Department of Biological Sciences, KAIST, Korea) for providing the *som-3* mutant and *SOM* overexpression lines, and the Nottingham Arabidopsis Stock Centre (NASC) for providing *erf*, *pif1-1*, and *abi5-8* mutant seeds.

## Author contributions

Conceptualisation: D.J.S., Z.L., M.S., A.H. Investigation: Z.L., D.J.S., E.v.R. Visualisation: Z.L. Writing – original draft: Z.L., M.S., A.H. Writing – review and editing: Z.L., D.J.S., E.v.R., M.S., A.H. Project administration: A.H.; Funding acquisition: Z.L., M.S., A.H.

## Funding

## Competing interests

The authors declare no competing interests.
