## [Peer Review File · Nature Communications]

The phytochrome interacting proteins ERF55 and ERF58 repress light-induced seed germination in *Arabidopsis thaliana*REVIEWER COMMENTS

Reviewer #1 (Remarks to the Author):

This article authored by Li et al described the role of two AP2/ERF-type transcription factors ERF55 and ERF58 in regulating light-mediated seed germination. They obtained the following results. (1) ERF58 interacts with phytochrome A (phyA) and phyB in vitro and in vivo via their Pfr form. (2) ERF55 and ERF58 negatively regulate phytochrome-mediated seed germination. (3) ERF58 binds directly to the promoters of PIF1 and SOM and induces their expression. (4) PhyA and phyB repress ERF55 and ERF58 gene expression and inhibit the DNA-binding activity of ERF55 and ERF58. (5) ERF58 directly binds to ABA2 (a ABA metabolic gene), ABI5 (a ABA signaling component), and GA2ox4 (a GA biosynthesis gene) to induce their expression. (6) ABA and GA regulate the expression of ERF55 and ERF58. Finally, they propose a working model to show the role of these two ERF factors.

Overall, the experiments were well designed and the manuscript was written in a clear manner. This study gains new factors and provide new insight into the understanding of light-mediated seed germination.

Here are some comments to improve the manuscript.

1. The authors revealed that ERF55 and ERF58 directly binds to PIF1, SOM, ABI5 and promote their expression and they showed that overexpression of either PIF1, SOM, or ABI5 complemented the seed germination phenotype of erf55erf58. As PIF1-SOM—ABI5 forms a signaling pathway in regulating seed germination (Figure 9 model), then why do ERF55 and ERF58 directly regulate the expression of these different signaling factors?
2. ERF55 and ERF58 additively regulate seed germination and the authors proposed in the model that they form dimers with phytochromes. I am wondering whether ERF55 and ERF58 could interact to form heterodimers and act with phytochrome together. If it was true, then the model could be revised.
3. To demonstrated that phyA and phyB repress ERF55 and ERF58 expression, it is necessary to include phyA and phyB mutants in the experiments in Fig S6.
4. There are a number of biosynthetic genes of the ABA and GA pathways, please explain the rationale why only a small set of genes was examined in the study.
5. Figures 5, 6, and 7 could be combined into one figure.
6. Statistical analysis should be applied to the seed germination assays and luciferase activity assays in many figures.
7. In Fig S1, a phylogenetic tree of the ERF A6 subfamily (ERF53-62) can be provided so that readers may understand the relationship between their evolution and functon(interaction with phys).

8. Are erf58-1 and erf58-2 loss-of-function mutants? Fig S2 shows the reduced expression level of ERF58 in these mutants.
9. The titles of Fig S7 and S8 can be modified to “ERF58 binds to PIF1/SOM promoter”
10. Provide an allele number for erf55, e.g erf55-1.

Reviewer #2 (Remarks to the Author):

Li et al. report that phytochrome-interacting ERF55 and ERF58 repress light regulated seed germination in Arabidopsis. They identified ERF55 and ERF58 interacted with light activated PhyA and PhyB. Phytochrome interactions block its DNA binding to promoters of PIF1 and SOM, well characterized regulators of phytochrome-mediated seed germination. ERF55 and ERF58 also directly bind to promoters of genes encoding ABA and GA metabolism. ABA and GA also regulate expression of ERF55 and ERF58. The authors propose this self-enhancing signaling loop regulates germination under permissive light conditions.

This work investigated an important biological topics that might attract readers who study plant signaling. Light signaling in plants is unique when compared with other organisms, thus this work might also provide insights on non-plant biologists. This work includes some interesting findings. In particular, finding of novel phytochrome interacting TFs and its mode of action is of interest. Methodology seems to be sound. However, I believe that current data is not sufficient to add on the current our knowledge.

Main concerns

1. Hormonal regulation of ERFs

Light regulated ABA/GA metabolism is well characterized in Arabidopsis. Particular sets of genes encoding late limiting enzymes show phytochrome-regulated expression. Those include NCED6, GA3ox and GA2ox2 etc. This work reports a different set of hormone metabolism genes are regulated by ERFs. As for the pif1, genes that show light-regulated expression in wt are misexpressed. Also, mutants of nced6, ga3ox show defects in light regulated germination. I wonder if NCED6 GA3ox GA2ox2 are not phytochrome-regulated in their conditions. Otherwise, are ABA2 and GA2ox4, not NCED6 etc, the primary phytochrome-regulated ABA/GA genes in this condition? Based on the model, ERF55 and ERF58 act through both PIF1 dependent and independent pathways, ABA/GA genes under the control of ERFs may suggest the PIF1-dependent ERF pathway is not acting well. Mutant phenotypes in Fig 2 are clear,

so I believe the authors' claim might be right. However, current data is insufficient to convince readers that proposed mechanisms is the base of the phenotype.

I don't see in the previous other papers that transcriptional regulation of ABA2 significant impact endogenous ABA levels. It has some effects to alter the ABA levels when rate-limiting NCED is fully activated. I wonder if a specific mutation in DRE of ABA2 promoter indeed alters the endogenous ABA levels.

I note that NCED9 is the primary ABA biosynthesis gene in high temperature-regulated seed germination, another phytochrome regulated process. I wonder if ERFs regulate a different phytochrome-regulated process, known as temperature signaling.

The role of PIF1 in light regulated germination is well understood. PIF1 is thought to be a light primary response regulator, thus PIF1 protein is already present when phytochrome is activated by light. I think what the authors should convince readers for the importance of PIF1 gene expression in the context of light-mediated germination. Is ERF-regulated PIF1 expression important during seed development prior to phytochrome action in germination? Otherwise, are ERFs involved in maintenance phase rather than induction?

In general, I have impression on this manuscript that is mechanism-oriented with missing sufficient biological context.

Reviewer #3 (Remarks to the Author):

The quality of the data — whether they are technically sound, obtained with appropriate techniques, analysed and interpreted carefully, and presented in sufficient detail.

This is a comprehensive and compelling body of work produced by a well respected PI in the field of plant light perception and the physiological consequences of that perception. The results are a tour de force of the latest techniques to ascertain an entire course of impact for two ERF transcription factors. The identity of important targets to which the ERFs bind are revealed, the consequence of binding...or not binding predicated on light regime and activation of phytochromes A or B on transcript abundance is measured; protein amounts determined; the influence this has on the ABA/GA balance elucidated; and

the whole tied in nicely with the completion or not, of seed germination. Certainly this manuscript presents a complete story and the data are of high quality.

The level of support for the conclusions — whether sufficiently strong evidence is provided for the authors' claims and all appropriate controls have been included.

In only a few instances have the authors extended their interpretation of their results outside the confines of the data collected. These instances have been pointed out in the text. For the major revelations they are reporting, they have ample evidence and have included all possible controls. My only major quibble is with their model where they show direct repressive effects (presumably through transcriptional down regulation) of ABA catabolic and GA anabolic enzymes when they have already pointed out that the ERF family of transcription factors are not known to possess repressive capacity.

The potential significance of the results — whether these results will be important to the field and advance understanding in a way that will move the field forward. (Note that posting of preprints and/or conference proceedings does not compromise novelty.)

There have been a wide array of manuscripts reporting on the influence photons have on all aspects of a plant's life cycle. However, certain major players, such as the PHYTOCHROME INTERACTING FACTORS have had prominence for a variety of reasons. The PIFs (specifically PIF1) is influential in determining the completion of seed germination in the dark, largely through alterations in ABA/GA titers, the so called hormone balance theory of seed germination. And yet PIF1 does not seem to directly influence the transcription of genes whose encoded proteins are capable of altering the titer of either GA or ABA. The authors now report that, in tandem (but not as a heterodimer), ERF58 and 55 are capable of directly interacting with the genes encoding proteins capable of altering the ABA/GA ratio in germinating seeds leading either to the completion of seed germination or its prolongation.

General Comments:

Provide the precise mutant line identifications, SALK #, SAIL#, GABI-CAT# for all mutant lines in materials. Do not refer me to earlier publications for this.

Perform a pixel density analysis and provide the graphs for all western blot images. If you have multiple replications, provide this volumetric assay graph with statistics.

Abandon the sloppy language! Avoid stating "seed germination" when you are referring to the "completion of germination". If that is too ponderous to write used instead "embryo protrusion".

You did not measure the "germination rate"! What is it doing in your text?? You may as well entitle your excellent manuscript, "The novel -chrome interacting proteins...." meaning cryptochrome or phytochrome, you know, one of the -chromes! You would be outraged at such unprofessional disregard for an established terminology! Do not perpetuate such loose prose, write what you mean, mean what you write. You have cited Bewley et als book, use it!

Provide statistics to distinguish among the various genotypes final cumulative germination percentages e.g. Supplemental figure 9.

The language suggesting that genes can somehow metabolize ABA or perform other enzymatic functions is infuriating coming from someone as erudite as this primary author! This, along with the loose language concerning the phenomenon under investigation (seed germination) MUST be rectified prior to publication of this work. On this point the editor must hold firm. Nature Communications is a paradigm of scientific communication due to its stringency. If this reputation is to be upheld, then there is to be no equivocation on the proper use of established terms and definitions!

Address the supershift for ERF58 in the EMSAs. Is there reason to believe this is real i.e. ERF58 homodimerization on target promoters?

Otherwise, so very well done! Kudos!

RESPONSE TO THE REVIEWERS

REVIEWER #1

Reviewer comment:

1. The authors revealed that ERF55 and ERF58 directly binds to PIF1, SOM, ABI5 and promote their expression and they showed that overexpression of either PIF1, SOM, or ABI5 complemented the seed germination phenotype of *erf55erf58*. As PIF1-SOM—ABI5 forms a signaling pathway in regulating seed germination (Figure 9 model), then why do ERF55 and ERF58 directly regulate the expression of these different signaling factors?

Response:

We agree that shutting off one factor in a linear pathway is sufficient to shut off the whole pathway. However, when only reducing the activity of one factor (e.g. by reducing but not fully inhibiting its expression), reducing the activity of another factor in the pathway would have an additional effect. Moreover, when considering a network instead of a linear pathway, a signal through the network may be rerouted if only one factor is shut off and it might be necessary to reduce the activity of several factors to obtain an efficient regulation of a specific response (Van den Broeck et al., 2020).

Seed germination is a highly complex trait controlled by a multitude of endogenous and environmental signals that need to be integrated to achieve an appropriate response. Signal integration depends on a network and is hardly possible with a simple linear signalling pathway. PIF1, SOM, and ABI5 are important nodes in this network but the view that they form a linear signalling pathway does not reflect the situation *in planta* (Bassel et al., 2011). Therefore, regulation of several nodes in the network by ERF55/ERF58 may be key to efficient regulation of the completion of germination in response to light.

Reviewer comment:

2. ERF55 and ERF58 additively regulate seed germination and the authors proposed in the model that they form dimers with phytochromes. I am wondering whether ERF55 and ERF58 could interact to form heterodimers and act with phytochrome together. If it was true, then the model could be revised.

Response:

This is a very good point. While dimerisation of bHLH transcription factors is well documented in the literature, we found only three publications reporting on the dimerisation of AP2/ERFs. One publication has shown that AP2/ERFs belonging to the cytokinin response factor (CRF) subfamily homo- and heterodimerise through a motif exclusively present in members of the CRF subfamily

but not in other AP2/ERFs (Striberny et al., 2017). The other two publications show that different members of subfamily B-3 can form homo- and/or heterodimers but the domain/motif for dimerisation is still unknown (Son et al., 2012; Huang et al., 2021). Thus, it appears that dimerisation of AP2/ERF transcription factors has not been investigated in much detail. Preliminary data from initial experiments that we performed indicate that ERF55/ERF58 possibly can form homo- and heterodimers. However, we prefer not to include this data in this manuscript for several reasons: first, the data are too preliminary and need further verification; second, knowing whether ERF55/ERF58 form dimers or not is not required for the conclusion that ERF55/ERF58 bind to phytochromes and to specific motifs in the promoter of direct target genes, i.e. for the conclusions of the manuscript; third, including additional data into the manuscript would go beyond the scope and blur the focus of the manuscript, which is on the initial characterisation of the function of ERF55/ERF58 in phytochrome signalling. However, we agree with the reviewer that dimerisation of AP2/ERFs is an important and interesting aspect and therefore we want to address it in future experiments.

Reviewer comment:

3. To demonstrated that phyA and phyB repress ERF55 and ERF58 expression, it is necessary to include phyA and phyB mutants in the experiments in Fig S6.

Response:

We fully agree with the reviewer and performed the experiment. New data included in the revised version of Supplementary Figure 6 show that downregulation of *ERF55* and *ERF58* transcript levels under phyA-ON or phyB-ON conditions is largely abolished in the respective mutant background (i.e. *phyA-211* or *phyB-9*), supporting our conclusion that phyA and phyB regulate the expression of *ERF55* and *ERF58*.

Reviewer comment:

4. There are a number of biosynthetic genes of the ABA and GA pathways, please explain the rationale why only a small set of genes was examined in the study.

Response:

The set of genes encoding ABA or GA metabolic enzymes for which we quantified transcript levels is based on the set investigated in other publications on seed germination (Barros-Galvão et al., 2020; Jiang et al., 2016; Kim et al., 2008; Lee et al., 2012; Li et al., 2019; Lim et al., 2013; Oh et al., 2006, 2007; Yang et al., 2020). We quantified transcript levels of most genes that are commonly analysed except for *NCED9* and *AAO3*. Therefore, in the revised version of the manuscript, we added new data on *NCED9* and *AAO3* transcript levels in Col-0 vs. *erf55 erf58* under phyA-ON/OFF and phyB-ON/OFF conditions (see Supplementary Figure 12). The data are in line with our conclusion that ERF55 and ERF58 enhance the expression of genes encoding ABA anabolic or GA catabolic enzymes, while they repress the expression of genes that encode ABA catabolic or GA anabolic enzymes. Overall, together with the new data on *NCED9* and *AAO3*, the set of genes that we tested largely corresponds to the set of genes tested in other studies on seed germination.

Reviewer comment:

5. Figures 5, 6, and 7 could be combined into one figure.

Response:

We agree with the reviewer that Figures 5, 6, and 7 address the same questions for different genes and could therefore be combined. However, combining the three figures would result in a figure with more than 10 panels that would not fit on one page. Therefore, in terms of layout, it may not be possible to combine them into one figure. In the revised version of the manuscript, we have not combined the three figures into one, but are happy to do so if the layout editor confirms that this is possible.

Reviewer comment:

6. Statistical analysis should be applied to the seed germination assays and luciferase activity assays in many figures.

Response:

We agree with the reviewer and performed statistical analysis on datasets from qPCR, ChIP-qPCR, luciferase activity assays, and germination experiments. Details on statistical tests are included in the figure legends and the source data file.

Reviewer comment:

7. In Fig S1, a phylogenetic tree of the ERF A6 subfamily (ERF53-62) can be provided so that readers may understand the relationship between their evolution and function(interaction with phys).

Response:

We fully understand the reviewer's request, but prefer not to show a phylogenetic tree for the following reason. We have used different methods to calculate phylogenetic trees of the AP2/ERF family and the ERF A6 subfamily. However, since ERFs are very short proteins and similarity is limited to the AP2 domain, it is difficult to obtain consistent results (i.e. different methods lead to different topologies of the tree); moreover, the support values for the nodes are generally low. This problem is reflected in different publications showing phylogenetic trees for the ERF family or ERF subfamilies that have conflicting topologies (Feng et al., 2020; Jin et al., 2017; Nakano et al., 2006). Under these circumstances, we believe it is better not to include a phylogenetic tree than to include a phylogenetic tree that does not meet adequate quality standards.

Reviewer comment:

8. Are *erf58-1* and *erf58-2* loss-of-function mutants? Fig S2 shows the reduced expression level of *ERF58* in these mutants.

Response:

According to T-DNA Express (<http://signal.salk.edu/cgi-bin/tdnaexpress>), both *erf58-1* and *erf58-2* contain a T-DNA insertion in the coding sequence of *ERF58*, which we confirmed by genotyping. Nevertheless, in qPCR experiments with two independent primer pairs, we can detect low *ERF58* transcript levels in both alleles with both primer pairs (data for one primer pair is shown in

Supplementary Figure 2) and therefore we do not want to fully exclude the possibility that *erf58- 1* and *erf58- 2* could be knock-down rather than knock-out alleles. Despite this uncertainty, we decided to use the *erf58- 1* and *erf58- 2* alleles, because no other *erf58* mutant alleles are available. It is important to note that both *erf58* alleles have the same phenotype and we can complement the *erf58* mutant by expression of an *pERF58:HA-YFP-ERF58* transgene, confirming that the phenotype of the *erf58* mutant alleles is indeed caused by the lack/strongly reduced levels of *ERF58* transcript.

Reviewer comment:

9. The titles of Fig S7 and S8 can be modified to “ERF58 binds to PIF1/SOM promoter”

Response:

We agree that binding of ERF55/ERF58 to the *PIF1* or *SOM* promoter in the transactivation assays is the most likely explanation for the positive effect of ERF55/ERF58 on the expression of the luciferase reporter. However, formally, the assay only shows that expression of luciferase is upregulated (i.e. the promoter driving luciferase expression is activated), not that ERF55/ERF58 bind to the promoter. Therefore, we prefer to write "activate the promoter" rather than "bind to the promoter".

Reviewer comment:

10. Provide an allele number for *erf55*, e.g *erf55-1*.

Response:

That is a very good point. We did not find any allele name in the literature for the *ERF55* T-DNA insertion line that we used in our study, so we assigned it the allele number "1", i.e. *erf55- 1*. We replaced *erf55* with *erf55- 1* throughout the text and figures.

REVIEWER #2

Reviewer comment:

Light regulated ABA/GA metabolism is well characterized in Arabidopsis. Particular sets of genes encoding late limiting enzymes show phytochrome-regulated expression. Those include *NCED6*, *GA3ox* and *GA2ox2* etc. This work reports a different set of hormone metabolism genes are regulated by ERFs. As for the *pif1*, genes that show light-regulated expression in wt are misexpressed. Also, mutants of *nced6*, *ga3ox* show defects in light regulated germination. I wonder if *NCED6* *GA3ox* *GA2ox2* are not phytochrome-regulated in their conditions. Otherwise, are *ABA2* and *GA2ox4*, not *NCED6* etc, the primary phytochrome-regulated ABA/GA genes in this condition?

Response:

We quantified the expression levels of *ABAI*, *ABA2*, *NCED6*, *CYP707A2*, *GA2ox2*, *GA2ox4*, *GA3ox1*, and *GA3ox2*; in addition, in the revised version of the manuscript, we added new data for

NCED9 and *AAO3* (Supplementary Figure 12). These are genes whose expression has been quantified in numerous studies on seed germination (Barros-Galvão et al., 2020; Jiang et al., 2016; Kim et al., 2008; Lee et al., 2012; Li et al., 2019; Lim et al., 2013; Oh et al., 2006; Yang et al., 2020). All these genes, including *NCED6*, *GA3ox*, and *GA2ox2*, are differentially regulated in phyA-ON vs. phyA-OFF or phyB-ON vs. phyB-OFF conditions and differential regulation is reduced or fully lost in *erf55 erf58*, suggesting that ERF55/ERF58 are required for the light regulation of these genes. For a subset of these genes, in particular genes that contain a DRE element in the promoter, we tested direct binding of ERF55/ERF58 to the promoter region containing the DRE element using EMSA. In the revised version of the manuscript, we present new data showing that ERF55/ERF58 also bind to promoter fragments of *NCED9* and *AAO3* containing a DRE element (Supplementary Figure 14). For *ABA2* and *GA2ox4*, we also confirmed binding of ERF58 to the promoter *in planta* using ChIP-qPCR. Thus, overall, we can safely conclude that ERF55 and ERF58 play a role in light-regulation of a standard set of genes encoding GA or ABA metabolic enzymes; part of these genes are direct targets of ERF55/ERF58, while others may be indirectly regulated.

To our knowledge, neither PIF1 nor SOM have been shown to directly bind to the promoter of genes encoding GA or ABA metabolic enzymes (Kim et al., 2008; Oh et al., 2004, 2009). In contrast, ERF55/ERF58 directly bind to the promoter of a subset of these genes and therefore our manuscript provides novel insight into how these genes are regulated. In addition, several genes that are regulated by ERF55/ERF58, but for which we have not yet tested binding of ERF55/ERF58 to the promoter, contain DRE elements in their promoter, suggesting they may be direct targets of ERF55/ERF58 as well.

Reviewer comment:

Based on the model, ERF55 and ERF58 act through both PIF1 dependent and independent pathways, ABA/GA genes under the control of ERFs may suggest the PIF1-dependent ERF pathway is not acting well. Mutant phenotypes in Fig 2 are clear, so I believe the authors' claim might be right. However, current data is insufficient to convince readers that proposed mechanisms is the base of the phenotype.

Response:

Using phyA/phyB-specific light treatments, we show that ERF55/ERF58 are required for proper light-regulation of several genes encoding ABA or GA metabolic enzymes, of *SOM*, a key factor in regulation of germination completion, as well as of *ABI5*, an essential component of ABA downstream signalling. As highlighted by reviewer #3, this clearly provides a link between light, ERF55/ERF58, and the completion of seed germination. We also show that transcript levels of *ERF55* and *ERF58* are regulated by light and that this regulation strongly depends on phyA and phyB (new data shown in Supplemental Figure 6). Regulation of ERF55/ERF58 at the transcript level is therefore one potential mechanisms by which phyA/phyB can regulate the completion of seed germination. In addition, using EMSAs we also observed that phyA prevents binding of ERF55/ERF58 to the promoter of several direct target genes. To further support this conclusion, we added new data (Supplemental Figure 14) in the revised version, showing that light-activated phyA also prevents binding of ERF55 and ERF58 to promoter fragments of *NCED9* and *AAO3*. Finally,

we show in our manuscript that a short light treatment activating or inactivating phyB prevents or enhances the association of ERF58 with the promoter of selected target genes.

It is true that we cannot decide to what degree transcriptional regulation of *ERF55/ERF58* and regulation of association of ERF55/ERF58 with target promoters contribute to the final response. However, we want to point out that this is not an ERF55/ERF58-specific problem, rather than a general problem for many transcription factors. For instance, the function of PIFs in light signalling, including the role of PIF1 in regulation of germination completion, has been known for almost two decades and numerous research groups have addressed the question how – in molecular terms – phytochromes control the activity of PIFs. The result is that we know today that phytochromes promote the phosphorylation of PIF and thereby enhance their degradation and that phytochromes bind to PIFs and prevent their association with target promoters (Al-Sady et al., 2006; Park et al., 2012). Yet, it is still unknown to what degree the two mechanisms contribute to the final response. This is similar to ERF55/ERF58 for which we show that phytochromes regulate the transcript levels and the association with target promoter but cannot exactly quantify the contribution of each mechanism.

Reviewer comment:

I don't see in the previous other papers that transcriptional regulation of *ABA2* significant impact endogenous ABA levels. It has some effects to alter the ABA levels when rate-limiting *NCED* is fully activated. I wonder if a specific mutation in DRE of *ABA2* promoter indeed alters the endogenous ABA levels.

Response:

ABA2 is a single copy gene that is required for ABA biosynthesis (Chauffour et al., 2019; Okamoto et al., 2010). There is a recent publication showing that the expression of *ABA2* is repressed by exogenously applied BR, while it is upregulated upon treatment with brassinazole as well as in *det2* and *cyp85a1 cyp85a2* mutant background. Repression or upregulation of *ABA2* correlated with reduced or increased ABA levels, respectively, showing that transcriptional regulation of *ABA2* can have an effect on levels of endogenous ABA (Moon et al., 2021). *BZR1* regulates BR signalling and it has been demonstrated by EMSA and ChIP assays that *BZR1* directly binds to the *ABA2* promoter (Moon et al., 2021).

We have shown i) that ERF55/ERF58 regulate the expression of *ABA2*, ii) that ERF55/ERF58 bind to an *ABA2* promoter fragment containing a DRE element, while binding is abolished by mutating this motif, iii) that light-activated phyA prevents the association of ERF55/ERF58 with the DRE containing *ABA2* promoter fragment, and iv) that light treatments leading to low levels of active phyB promote the association of ERF58 with the *ABA2* promoter *in planta*, while high levels of active phyB promote the dissociation of ERF58 from the promoter of *ABA2*. Furthermore, we show that functional ERF55/ERF58 lead to high levels of endogenous ABA, while levels of endogenous ABA are reduced in the *erf55 erf58* mutant. This data is consistent with a model in which ERF55/ERF58 regulate transcript levels of *ABA2* and thereby can affect the endogenous levels of ABA. However, we agree with the reviewer that also regulation of *NCEDs* is important to regulate the levels of endogenous ABA. Data presented in the manuscript indeed show that *NCED6* transcript levels are downregulated in *erf55 erf58* mutant seed compared to wildtype seeds and in the revised version of the manuscript we provide new data in

Supplementary Figures 12c, h, and 14a showing that i) also transcript levels of *NCED9* are lower in seeds of the *erf55 erf58* mutant than in the wildtype, ii) that both ERF55 and ERF58 associate with an *NCED9* promoter fragment containing a DRE element, and iii) that light-activated phyA prevents the association of ERF55/ERF58 with this *NCED9* promoter fragment. In addition, in the revised version of the manuscript we include new data (Supplementary Figures 12d, i, and 14b), showing that *AAO3*, another gene encoding an ABA anabolic enzyme, is regulated in a similar manner as *ABA1*, *ABA2*, *NCED6*, and *NCED9*.

Reviewer comment:

I note that *NCED9* is the primary ABA biosynthesis gene in high temperature-regulated seed germination, another phytochrome regulated process. I wonder if ERFs regulate a different phytochrome-regulated process, known as temperature signaling.

Response:

In the revised version of the manuscript, we included new data showing that *NCED9* expression is regulated by ERF55/ERF58 and phytochromes (see response to previous comment). Thus, the comment by reviewer #2 is very interesting but addressing this question would go beyond the scope of the current manuscript. Instead, we plan to test a potential function of ERF55/ERF58 in temperature-regulation of the completion of seed germination in future work.

Reviewer comment:

The role of PIF1 in light regulated germination is well understood. PIF1 is thought to be a light primary response regulator, thus PIF1 protein is already present when phytochrome is activated by light. I think what the authors should convince readers for the importance of PIF1 gene expression in the context of light-mediated germination. Is ERF-regulated PIF1 expression important during seed development prior to phytochrome action in germination? Otherwise, are ERFs involved in maintenance phase rather than induction?

Response:

We found that ERF55/ERF58 associate with the promoter of *PIF1* in EMSA and that light-activated phyA prevents this association. Furthermore, in ChIP-qPCR experiments, ERF58 associates with the *PIF1* promoter under conditions where phyB is inactive while activation of phyB promotes dissociation from the *PIF1* promoter. Consistent with the idea that ERF55/ERF58 contribute to the regulation of *PIF1* expression, we found that *PIF1* transcript levels are lower in *erf55 erf58* than in the wildtype. However, the lack of functional ERF55 and ERF58 has a stronger effect on many other genes for which we quantified transcript levels than on *PIF1*, including *SOM* and most genes encoding ABA or GA metabolic enzymes. Thus, regulating ABA and possibly GA levels through regulation of genes encoding ABA or GA metabolic enzymes (either directly or through *SOM*) might be the dominant mechanisms by which ERF55/ERF58 contribute to regulation of germination completion.

It is a very interesting idea that ERF55/ERF58-dependent regulation of *PIF1* expression could play a role during seed development but addressing this question goes beyond the scope of this manuscript and we plan to address it in future work.

Reviewer comment:

In general, I have impression on this manuscript that is mechanism-oriented with missing sufficient biological context.

Response:

We explained the biological context of our work in the introduction and discussion by pointing out the critical role of proper regulation of germination completion for the fitness of a plant population and for agriculture. In the result section, we show that ERF55/ERF58 play a role in this process by contributing to the regulation of germination completion depending on the light environment. We think this provides a clear biological background to our work, but we are happy to expand the introduction or discussion if necessary.

REVIEWER #3**Reviewer comment:**

- In only a few instances have the authors extended their interpretation of their results outside the confines of the data collected. These instances have been pointed out in the text. For the major revelations they are reporting, they have ample evidence and have included all possible controls. My only major quibble is with their model where they show direct repressive effects (presumably through transcriptional down regulation) of ABA catabolic and GA anabolic enzymes when they have already pointed out that the ERF family of transcription factors are not known to possess repressive capacity.
- This is not consistent with what you are showing in your model with a direct line of repression from the ERFs to ABA catabolic and GA anabolicenzymes.

Response:

We show that ERF55/ERF58 downregulate transcript levels of selected genes encoding ABA catabolic or GA anabolic enzymes. However, this is not necessarily a direct effect of ERF55/ERF58 and the promoter of most of these genes does not contain a DRE element. Thus, these genes are possibly indirectly regulated by ERF55/ERF58. The expression of these genes is also regulated by SOM and it is possible that the effect of ERF55/ERF58 on *SOM* expression is responsible for ERF55/ERF58-mediated downregulation of genes encoding ABA catabolic or GA anabolic genes. All genes that we identified as direct targets of ERF55/ERF58 using EMSAs and/or ChIP-qPCR are i) upregulated by ERF55/ERF58 (downregulated in *erf58 erf58* compared to WT), and ii) encode proteins that lead to repression of the completion of seed germination. We think that this was not sufficiently clear in our model and therefore revised Figure 9. In the revised version of Figure 9, we clearly distinguish between direct and indirect targets of ERF55/ERF58. In addition, we also added a few lines on this in the discussion.

Reviewer comment:

Provide the precise mutant line identifications, SALK #, SAIL#, GABI-CAT# for all mutant lines in materials. Do not refer me to earlier publications for this.

Response:

This is a good point and we provided stock numbers for all mutant lines in the revised version of the manuscript (both in the Method section as well as in Supplementary Table 3).

Reviewer comment:

Abandon the sloppy language! Avoid stating "seed germination" when you are referring to the "completion of germination". If that is too ponderous to write used instead "embryo protrusion".

Response:

We have followed the language commonly used by many photobiologists and were not aware of how inaccurate and inappropriate it is. Therefore, we are very grateful to reviewer #3, obviously an expert in seed biology, for taking so much time to carefully check our manuscript and making suggestions for the correct wording. We accepted all the changes he/she suggested in the annotated pdf and corrected some additional cases where "germination" was used in an appropriate way.

Reviewer comment:

- Provide statistics to distinguish among the various genotypes final cumulative germination percentages e.g. Supplemental figure 9.
- Statistics are required to indicated the truly significant differences among the various genotypes. This goes for most of your comparisons of the final cumulative percentage germination.

Response:

We agree with the reviewer and performed statistical analysis on datasets from qPCR, ChIP-qPCR, luciferase activity assays, and germination experiments. Details on statistical tests are included in the figure legends and the source data file.

Reviewer comment:

The language suggesting that genes can somehow metabolize ABA or perform other enzymatic functions is infuriating coming from someone as erudite as this primary author! This, along with the loose language concerning the phenomenon under investigation (seed germination) MUST be rectified prior to publication of this work. On this point the editor must hold firm. Nature Communications is a paradigm of scientific communication due to its stringency. If this reputation is to be upheld, then there is to be no equivocation on the proper use of established terms and definitions!

Response:

We fully agree with the reviewer that a gene encodes a protein that has a specific function, i.e. the protein performs the function, not the gene, and therefore there are no metabolic genes. We greatly appreciate the efforts by reviewer #3 to improve proper and precise use of language in our manuscript and incorporated all his/her suggestions.

Reviewer comment:

- Address the supershift for ERF58 in the EMSAs. Is there reason to believe this is real i.e. ERF58 homodimerization on target promoters?

- Supershift? Care to say anything about ERF58:ERF58 dimerization?
- Very probably a super shift for ERF58!! Any evidence of self binding of the protein?
- Less obvious but still a supershift evident. Is this just excess amounts of ERF58 in the assays or is this real? Any evidence either way?
- ERF58 Supershift evident.

Response:

This is an interesting observation. We have at least three replicates for all EMSAs (for some up to six replicates) and we checked them carefully. The potential supershift can be observed in some replicates but clearly not in all. As described in the response to question 2 by reviewer #1, we have preliminary data showing that ERF55 and ERF58 may form homo- and heterodimers, but further experiments are required for final conclusions. Nevertheless, we used ERF55/ERF58 fusion proteins that differ in size for EMSAs (e.g. MBP-ERF58/GST-ERF58; MBP-ERF58 and GST-ERF58 for comparison). For binding of MBP-ERF58/GST-ERF58 heterodimers we would expect a different shift compared to binding of MBP-ERF58 or GST-ERF58 homodimers, if ERF58 forms dimers. However, we could not observe a clear and reproducible difference for the different combinations. Overall, it is too preliminary to draw conclusions on whether ERF55/ERF58 form homo- and/or heterodimers and whether or not they bind to target promoters as homo- and/or heterodimers. Yet, this uncertainty in no way affects the conclusion that ERF55/ERF58 bind to target promoters in a phy-regulated manner and control the expression of genes involved in the regulation of germination completion.

It is an intriguing possibility that homo- and heterodimers of ERF55/ERF58 may have slightly different binding properties and regulate partially different sets of target genes. Therefore, in future work we want to investigate homo-/heterodimerisation and binding of homo-/heterodimers to target promoters.

Reviewer comment:

- For ALL western blots, quantify the pixel densities using a program for experimental and control and provide a graph of normalized protein abundances. Actin is too variable for an accurate assessment of variations in experimental protein abundances by eye.
- Provide volumetric analysis (pixel Densities of ERF protein abundance normalized to pixel densities of you Actin control for both of these blots). Actin varies too much to be able to assess this by eye.

Response:

We quantified signals on all western blot membranes as suggested by the reviewer. In the revised version of the figures, numbers below western blot membranes show pixel density for the band obtained with the respective antibody divided by the pixel density for the band obtained with the anti-actin antibody (i.e. signals were normalised for actin abundance).

Reviewer comment:

Please provide a cartoon of the promoters you have used here similar to the one you provide for the supplemental figures. Include the length from the ATG.

Response:

We have added a new panel in Figure 3 (Figure 3c) showing schematic drawings of the constructs used for the experiment in Figure 3d.

Reviewer comment:

Please consider relabeling this as phyB-ON; phyB-ON-OFF; phyB-OFF and; phyB-OFF-ON. I found the description of the phy-ON-FR as inactivating while phy-OFF-R was activating more difficult to follow than simply stating the final physiological status of the phytochromes would be.

i.e. your audience would have to know that phyA takes 12 hrs to come up from the commencement of seed germination while phyB is present in the mature, dry seed (Shinomura's work published in PNAS) to understand why the phy-ON-FR @ 6 h 40' does not allow sufficient phyA to stay in the Pfr form (it is not yet present).

Response:

We fully agree with the reviewer that the labelling is confusing and changed it in all figures and throughout the text.

Reviewer comment:

- Looking at your supplemental figure 11 b and c, third lane over + ERF58 + pPIF1 no shift in pPIF1! Fourth lane, what does "cold" signify? Fifth and sixth lanes, ERF58 present or not the phyA pPIF1 combination produce an EMSA shift. Explain this figure and its significant much better than you have!

- What about it?? "...to which the ERFs bind..."??? Explain this in this and supplemental 11 figure legends.

Response:

- We agree that Figure 11 is not sufficiently well explained. Figure 11b and c shows a Zinc blot for the EMSA samples, i.e. the signal is not due to the biotin-labelled probe but due to the fluorescence of a bilin/zinc complex. Bilin-linked polypeptides in the presence of Zinc-acetate form a complex that results in orange fluorescence when viewed under UV light. Here, we used this method to confirm that chromophore-bound phyA is present in the respective EMSA samples (Berkelman and Lagarias, 1986). EMSA samples were analysed on PAGE gels supplemented with Zinc-acetate and fluorescence signals were detected by exposing the PAGE gels to UV light. We added this explanation to the figure legend.

- Yes, "Cold" refers to an excess of an unlabelled DNA fragment containing the DRE motif to which ERF55/ERF58 bind. We added this explanation to the figure legends.

Reviewer comment:

You need to provide much more detail here. The take home from this is that no percentage of WT Col seeds can complete germination without at least 5uM GA. Look at your 0 uM paclobutrazol. Col WT do not complete germination on water??? Columbia seeds, even if harvested fresh, are not that dormant. So, maybe you have them all on 5uM Paclobutrazol and THEN feed them varying concentrations of GA to counteract the inhibitor? Say something here to explain the results you are showing. Consider splitting the graph in two if the continuous line from pac treated to GA treated is

causing me to misinterpret your results.

Response:

We apologise for the confusion caused by this figure. Yes, the reviewer is right, the figure shows two independent experiments in the same graph. We agree that this is confusing and in the revised version of the figure we show the data for the GA and the PAC experiment in separate panels/graphs. Figure 7a shows mean cumulative germination percentages of seeds treated with increasing GA3 concentrations incubated in the dark for seven days and Figure 7b shows mean cumulative germination percentages of seeds treated with increasing PAC concentrations exposed to white light for 6 hours followed by incubation in the dark for seven days. We added this explanation to the figure legend in the revised version of the manuscript.

Reviewer comment:

- Because there are many steps between the transcription of the genes you are indicating, production of protein, alteration of hormone concentrations in some instances, before we ever get to the completion of germination, may I suggest you use a dotted line instead of a solid line.
-"Germination completion"; this is what you measure, not the steps leading up to this event.

Response:

We revised Figure 9 and included the suggestions.

Reviewer comment:

What is delta N and delta C??

Response:

DeltaN and DeltaC refer to ERF58 truncations lacking the N-terminal or C-terminal domains, respectively. However, in Supplemental Table 1, we refer to these truncations as ERF58 1-171 and ERF58 78-261 (i.e. by giving the the first and last amino acid positions of the fragment). To be consistent, we changed the labels in Supplementary Figure 10 and now refer to ERF58 DeltaN and ERF58 DeltaC as ERF58 78-261 and ERF58 1-171, respectively. This is also explained in the figure legend of the revised version of Supplementary Figure 10.

References

- Al-Sady, B., Ni, W., Kircher, S., Schäfer, E., and Quail, P.H.** (2006). Photoactivated phytochrome induces rapid PIF3 phosphorylation prior to proteasome-mediated degradation. *Mol. Cell* **23**: 439–446.
- Barros-Galvão, T., Dave, A., Gilday, A.D., Harvey, D., Vaistij, F.E., and Graham, I.A.** (2020). ABA INSENSITIVE4 promotes rather than represses PHYA-dependent seed germination in *Arabidopsis thaliana*. *New Phytol.* **226**: 953–956.

- Bassel, G.W., Lan, H., Glaab, E., Gibbs, D.J., Gerjets, T., Krasnogor, N., Bonner, A.J., Holdsworth, M.J., and Provart, N.J.** (2011). Genome-wide network model capturing seed germination reveals coordinated regulation of plant cellular phase transitions. *Proc. Natl. Acad. Sci. U. S. A.* **108**: 9709–9714.
- Berkelman, T.R. and Lagarias, J.C.** (1986). Visualization of bilin-linked peptides and proteins in polyacrylamide gels. *Anal. Biochem.* **156**: 194–201.
- Chauffour, F. et al.** (2019). Multi-omics analysis reveals sequential roles for ABA during seed maturation. *Plant Physiol* **180**: 1198–1218.
- Feng, K., Hou, X.-L., Xing, G.-M., Liu, J.-X., Duan, A.-Q., Xu, Z.-S., Li, M.-Y., Zhuang, J., and Xiong, A.-S.** (2020). Advances in AP2/ERF super-family transcription factors in plant. *Crit. Rev. Biotechnol.* **40**: 750–776.
- Huang, J., Zhao, X., Bürger, M., Wang, Y., and Chory, J.** (2021). Two interacting ethylene response factors regulate heat stress response. *Plant Cell* **33**: 338–357.
- Jiang, Z., Xu, G., Jing, Y., Tang, W., and Lin, R.** (2016). Phytochrome B and REVEILLE1/2-mediated signalling controls seed dormancy and germination in *Arabidopsis*. *Nat. Commun.* **7**: 12377.
- Jin, J., Tian, F., Yang, D.-C., Meng, Y.-Q., Kong, L., Luo, J., and Gao, G.** (2017). PlantTFDB 4.0: toward a central hub for transcription factors and regulatory interactions in plants. *Nucleic Acids Res* **45**: D1040–D1045.
- Kim, D.H., Yamaguchi, S., Lim, S., Oh, E., Park, J., Hanada, A., Kamiya, Y., and Choi, G.** (2008). SOMNUS, a CCCH-type zinc finger protein in *Arabidopsis*, negatively regulates light-dependent seed germination downstream of PIL5. *Plant Cell* **20**: 1260–1277.
- Lee, K.P., Piskurewicz, U., Turečková, V., Carat, S., Chappuis, R., Strnad, M., Fankhauser, C., and Lopez-Molina, L.** (2012). Spatially and genetically distinct control of seed germination by phytochromes A and B. *Genes Dev.* **26**: 1984–1996.
- Li, C., Zheng, L., Wang, X., Hu, Z., Zheng, Y., Chen, Q., Hao, X., Xiao, X., Wang, X., Wang, G., and Zhang, Y.** (2019). Comprehensive expression analysis of *Arabidopsis GA2-oxidase* genes and their functional insights. *Plant Sci.* **285**: 1–13.
- Lim, S., Park, J., Lee, N., Jeong, J., Toh, S., Watanabe, A., Kim, J., Kang, H., Kim, D.H., Kawakami, N., and Choi, G.** (2013). ABA-INSENSITIVE 3, ABA-INSENSITIVE 5, and DELLAs interact to activate the expression of *SOMNUS* and other high-temperature-inducible genes in imbibed seeds in *Arabidopsis*. *Plant Cell* **25**: 4863–4878.
- Matsushita, T., Mochizuki, N., and Nagatani, A.** (2003). Dimers of the N-terminal domain of phytochrome B are functional in the nucleus. *Nature* **424**: 571–574.

- Moon, J., Park, C.-H., Son, S.-H., Youn, J.-H., and Kim, S.-K.** (2021). Endogenous level of abscisic acid down-regulated by brassinosteroids signaling via BZR1 to control the growth of *Arabidopsis thaliana*. *Plant Signal. Behav.* **16**: 1926130.
- Nakano, T., Suzuki, K., Fujimura, T., and Shinshi, H.** (2006). Genome-wide analysis of the ERF gene family in *Arabidopsis* and rice. *Plant Physiol.* **140**: 411–432.
- Oh, E., Kang, H., Yamaguchi, S., Park, J., Lee, D., Kamiya, Y., and Choi, G.** (2009). Genome-wide analysis of genes targeted by PHYTOCHROME INTERACTING FACTOR 3-LIKE5 during seed germination in *Arabidopsis*. *Plant Cell* **21**: 403–419.
- Oh, E., Kim, J., Park, E., Kim, J.-I., Kang, C., and Choi, G.** (2004). PIL5, a phytochrome-interacting basic helix-loop-helix protein, is a key negative regulator of seed germination in *Arabidopsis thaliana*. *Plant Cell* **16**: 3045–3058.
- Oh, E., Yamaguchi, S., Hu, J., Yusuke, J., Jung, B., Paik, I., Lee, H.-S., Sun, T., Kamiya, Y., and Choi, G.** (2007). PIL5, a phytochrome-interacting bHLH protein, regulates gibberellin responsiveness by binding directly to the *GAI* and *RGA* promoters in *Arabidopsis* seeds. *Plant Cell* **19**: 1192–1208.
- Oh, E., Yamaguchi, S., Kamiya, Y., Bae, G., Chung, W.-I., and Choi, G.** (2006). Light activates the degradation of PIL5 protein to promote seed germination through gibberellin in *Arabidopsis*. *Plant J.* **47**: 124–139.
- Okamoto, M. et al.** (2010). Genome-wide analysis of endogenous abscisic acid-mediated transcription in dry and imbibed seeds of *Arabidopsis* using tiling arrays. *Plant J.* **62**: 39–51.
- Osterlund, M.T., Hardtke, C.S., Wei, N., and Deng, X.W.** (2000). Targeted destabilization of HY5 during light-regulated development of *Arabidopsis*. *Nature* **405**: 462–466.
- Park, E., Park, J., Kim, J., Nagatani, A., Lagarias, J.C., and Choi, G.** (2012). Phytochrome B inhibits binding of PHYTOCHROME INTERACTING FACTORS to their target promoters. *Plant J.* **72**: 537–546.
- Son, G.H., Wan, J., Kim, H.J., Nguyen, X.C., Chung, W.S., Hong, J.C., and Stacey, G.** (2012). Ethylene-Responsive Element-Binding Factor 5, ERF5, is involved in chitin-induced innate immunity response. *Mol. Plant Microbe Interact.* **25**: 48–60.
- Striberny, B., Melton, A.E., Schwacke, R., Krause, K., Fischer, K., Goertzen, L.R., and Rashotte, A.M.** (2017). Cytokinin Response Factor 5 has transcriptional activity governed by its C-terminal domain. *Plant Signal. Behav.* **12**: e1276684.
- Van den Broeck, L., Gordon, M., Inzé, D., Williams, C., and Sozzani, R.** (2020). Gene regulatory network inference: connecting plant biology and mathematical modeling. *Front. Genet.* **11**: 457.

Yang, L., Jiang, Z., Jing, Y., and Lin, R. (2020). PIF1 and RVE1 form a transcriptional feedback loop to control light-mediated seed germination in Arabidopsis. *J. Integr. Plant Biol.* **62**: 1372–1384.

REVIEWERS' COMMENTS

Reviewer #1 (Remarks to the Author):

My previous comments have been mostly addressed.

Reviewer #2 (Remarks to the Author):

The revised manuscript improves the quality. The data is more compatible with previous knowledge of phytochrome-regulated seed germination in Arabidopsis. So, this work will add new knowledge on the known mechanisms. In particular, ABA and GA-related genes under the control of ERF55 and ERF58 are examined for ones controlled by PIF1 and SOM reported in previous papers. The manuscript is more balanced overall and more accessible for readers to follow the story.

I have some minor comments on the revised manuscript.

1. Thus, direct regulation of ABA2 expression by ERF55 and ERF58 could be of particular importance since ABA2 catalyzes an essential step in ABA biosynthesis and is not known to be controlled by PIF1 and SOM.

I recommend the author revise this sentence. It is true that ABA2 is indeed encoded as a single gene, and ALL ABA biosynthesis steps are essential. However, the essential step does not mean it is the regulatory step or rate-limiting step to alter ABA levels and trigger the physiological changes by altering ABA levels. In most of the cases, including seeds of Arabidopsis, the xanthoxin synthesis activity is abundant, so upregulation of this step has a minor/no visible role in increasing the ABA levels. This does not mean that ABA2 is not related to the regulation of ABA levels at all. It can occur under certain circumstances, including NCED is already activated (like a long-term response) or ABA2 is unusually down-regulated before the phenomenon you study. Otherwise, down-regulation of ABA2 could limit the ABA levels in theory. Whatever the cases, the current description looks like an inadequate argument.

I commented the previous manuscript is mechanism oriented with weak in context. This is because you demonstrate that ERF55/58 directly bind to the ABA2 promoter to regulate ABA2 expression (which is fine). My point is that you did not show whether the regulation of ABA2 expression indeed alters the

ABA levels or not in your context. It is possible, but it could occur only under certain circumstances (worth demonstrating if ABA2 regulation is associated with altering ABA levels in your context).

References 12-14 added in this sentence are not good choices to fully rationalize this sentence.

gibberellic acid (GA) Line 49 and keywords

I suggest you state gibberellin (GA) rather than gibberellic acid. For chemists, gibberellic acid is GA3, which is confusing. The word gibberellin is better for a wide range of readers.

Reviewer #3 (Remarks to the Author):

The novel phytochrome interacting proteins ERF55 and ERF58 repress light-induced seed germination in *Arabidopsis thaliana*.

Zenglin Li, David J. Sheerin, Edda von Roepenack-Lahaye, Mark Stahl, and Andreas Hiltbrunner

In the Abstract, the authors have separated the influence of GA and ABA (the hormone balance theory of seed germination, Karssen, Koornneef, and others) from the action of Phytochrome. In fact, I know of no instance that phytochrome imposes an influence on seed germination that does not ultimately work through the hormone balance of ABA and GA. PIF1, SPT, and SOM all promote (directly or indirectly) retention of biologically active ABA and can be argued to repress accumulation of biologically active GA which prevents the GID1:GA:RGL2 complex formation, which would otherwise make the DELLA protein susceptible to SLEEPY1 (SLY1: FBox protein) binding and DELLA destruction. Both ethylene and brassinosteroids have been demonstrated to reduce embryo sensitivity to ABA, indirectly stimulating the completion of germination (Steber, McCourt, and others).

“Seed germination is a critical step in the life cycle of plants controlled by the phytohormones abscisic acid (ABA) and gibberellin (GA), and in many cases also by phytochromes,…”

Last sentence of the Introduction, “...and thereby promotes the germination completion.”

Either, “...and thereby promotes germination completion.”

OR

“...and thereby promotes the completion of germination.”

Apart from these quibbles, all of my comments were addressed. I encourage publication of this work by Nature.

We thank the reviewers for their comments.

It appears that reviewer #2 refers to the original, rather than the revised, version of the manuscript. We addressed the points raised by reviewer #2 in the revised version – in part by changing the text and in part by performing additional experiments.

We thank reviewer #3 for the comments and changed the manuscript text accordingly.